# Construction and integration of three *de novo* Japanese human genome assemblies toward a population-specific reference

Jun Takayama [1,2,3], Shu Tadaka [2], Kenji Yano [2,3], Fumiki Katsuoka [1,2], Chinatsu Gocho[2], Takamitsu Funayama[2], Satoshi Makino[2], Yasunobu Okamura[1,2], Atsuo Kikuchi [4], Sachiyo Sugimoto[2], Junko Kawashima[2], Akihito Otsuki [2], Mika Sakurai-Yageta[2], Jun Yasuda[2,5], Shigeo Kure[2,4], Kengo Kinoshita [1,2,6 ✉], Masayuki Yamamoto [1,2 ✉] & Gen Tamiya[1,2,3,7 ✉]

The complete human genome sequence is used as a reference for next-generation sequencing analyses. However, some ethnic ancestries are under-represented in the reference genome (e.g., GRCh37) due to its bias toward European and African ancestries. Here, we perform de novo assembly of three Japanese male genomes using > 100× Pacific Biosciences long reads and Bionano Genomics optical maps per sample. We integrate the genomes using the major allele for consensus and anchor the scaffolds using genetic and radiation hybrid maps to reconstruct each chromosome. The resulting genome sequence, JG1, is contiguous, accurate, and carries the Japanese major allele at most loci. We adopt JG1 as the reference for confirmatory exome re-analyses of seven rare-disease Japanese families and find that re-analysis using JG1 reduces total candidate variant calls versus GRCh37 while retaining disease-causing variants. These results suggest that integrating multiple genomes from a single population can aid genome analyses of that population.

[1] Advanced Research Center for Innovations in Next-Generation Medicine, Tohoku University, 2-1, Seiryo-machi, Aoba-ku, Sendai, Miyagi 980-8573, Japan. [2] Tohoku Medical Megabank Organization, Tohoku University, 2-1, Seiryo-machi, Aoba-ku, Sendai, Miyagi 980-8573, Japan. [3] Statistical Genetics Team, RIKEN Center for Advanced Intelligence Project, Nihonbashi 1-chome Mitsui Building 15F, 1-4-1 Nihonbashi, Chuo-ku, Tokyo 103-0027, Japan. [4] Department of Pediatrics, Tohoku University Graduate School of Medicine, 2-1, Seiryo-machi, Aoba-ku, Sendai, Miyagi 980-8575, Japan. [5] Division of Molecular and Cellular Oncology, Miyagi Cancer Center Research Institute, 47-1, Nodayama, Medeshima-Shiode, Natori, Miyagi 981-1293, Japan. [6] Graduate School of Information Sciences, Tohoku University, 6-3-09 Aramaki Aza-Aoba, Aoba-ku, Sendai, Miyagi 980-8579, Japan. [7] Tohoku University Graduate School of Medicine, 2-1, Seiryo-machi, Aoba-ku, Sendai, Miyagi 980-8575, Japan. ✉email: kengo@ecei.tohoku.ac.jp; masiyamamoto@med.tohoku.ac.jp; gtamiya@megabank.tohoku.ac.jp

The complete human genome sequence[1,2] has been an invaluable resource for both basic research in human genetics and clinical diagnosis. The complete genome sequence—also called "the reference genome"—is currently used as a target for mapping the enormous number of short reads generated using major next-generation sequencing (NGS) techniques[3,4]. Because the short reads generated in NGS studies are approximately 100–300 bp in length, mapping them to the reference genome is an indispensable step for calling single nucleotide variants (SNVs) and short insertions and deletions (indels) in the sample individuals. The coordinate system of the reference genome is used for biological and medical annotations, such as the position or sequence of specific genes, or sites of causal variants associated with both rare and common diseases. Therefore, the reference genome is one of the most foundational resources in human genetics, and as such, it is maintained and continually updated by the Genome Reference Consortium (GRC). The latest and second-latest versions of the reference genome (GRCh38/hg38 and GRCh37/hg19, published in 2013 and in 2009, respectively) are nearly complete, and both are widely used for NGS analyses and genome annotations[5,6].

The reference genome was constructed using a hierarchical shotgun sequencing strategy in which fragmented genomic DNA segments cloned in bacterial (BAC) or P1-derived (PAC) artificial chromosome libraries were arranged into a correct physical map to guarantee that the reference genome was haploid (mosaic)[1]. The assembled contigs or scaffolds were then anchored on each chromosome using information from genetic and radiation hybrid (RH) maps, which have thousands to tens of thousands of sequence-tagged site (STS) markers in linkage groups (i.e., chromosomes). It should be noted that these genetic and RH maps are original information sources used to construct the reference genome and not derived from the reference genome itself.

Although the reference genome is a resource of unparalleled value, several of its characteristics are not ideal for application to NGS analyses, particularly for some populations[7]. For example, although the reference genome is constructed using genetic information from multiple donors, each clone comprising the resulting reference genome is derived from either haploid genome of a particular individual. As such, the reference genome inevitably harbors rare or even private variants. Over 90,000 rare variants were used as a reference allele including disease-susceptibility variants for thrombophilia and type 2 diabetes[8,9]. Inclusion of such variants in the reference can lead to erroneous and confusing results of short-read mapping or variant calling[9]. As NGS analyses typically assume that the reference allele is the ancestral, healthy, or major allele for any variable site, the inclusion of such rare alleles may also confuse subsequent interpretations.

Another possible problem associated with the reference genome is that the samples used for its construction are biased toward African and European ancestries. For example, > 70% of the reference genome is composed of a BAC library known as RP-11 (aliased RPCI-11)[1] from a donor with both African and European ancestry[10]. With the exception of one donor with an Asian background, all of the donors had a European background resulting in the composition of an Asian haplotype for 4.3% of the reference genome[1,10]. In addition, recent studies revealed a lack of (population-specific) sequences in the reference genome, and discovered thousands of structural variants (SVs) in worldwide samples[11–14]. These issues can also complicate short-read mapping and variant callings.

Several studies have examined ways to overcome the above-mentioned drawbacks to the reference genome. Dewey et al.[15] proposed modifying the reference genome by substituting its minor variants with the major variants from African, Asian, or European populations[15]. The resulting modified reference genome was better-suited for genome analyses of sample individuals with matched population backgrounds. Several studies[16–19] utilized a genome graph, which is an extended reference genome represented as a graph harboring known variants. Other studies have proposed the addition of sequences not included in the reference genome[11,12,20,21]. However, as these proposed adjustments are based largely on variants discovered using the reference genome itself, albeit only partially, in a circular fashion, some reference bias could remain.

One promising approach to address these problems is to construct reference genomes specific to ethnic populations of interest[22]. Although costly, highly contiguous de novo assembly—independent reconstruction—of the entire human genome is now feasible using, for example, Pacific Biosciences (PacBio) single molecule, real-time (SMRT) long reads (~10 kb in length) and Bionano Genomics (Bionano) optical mapping, which generates a high-resolution physical map[20,23–25]. Combining these approaches is known as 'hybrid scaffolding,' which is carried out in three steps: 1) PacBio long reads are de novo assembled to yield primary contigs; 2) Bionano optical maps are also de novo assembled (independent of the PacBio assembly) to yield genome maps; and 3) the PacBio-derived contigs are scaffolded by the Bionano genome maps. This strategy is analogous to the hierarchical shotgun sequencing strategy used in the Human Genome Project[1] with arrangements of long sequences from BAC/PAC on a physical map. Although assemblies generated in recent studies were highly contiguous and accurate, the assembled sequences were rarely anchored to a set of chromosomes (i.e., pseudo-molecules), thus making their use as references for NGS analyses impractical. Moreover, a single haploid assembly from a single individual cannot be used to solve the rare reference allele problem. A notable exception is the KOREF genome sequence[22], in which a Korean reference genome was constructed by de novo assembly of the genome sequence of a Korean individual, reconstructed as pseudo-molecules, and rare variants were substituted with short reads from 40 Korean individuals. However, the KOREF genome assembly was found to be less contiguous than long read-based assemblies because the primary sequencing platform was a short-read sequencer, and KOREF depended heavily on the reference genome because chromosome building was carried out by sequence-based alignment of scaffolds onto GRCh38.

In this work, using a hybrid scaffolding strategy, we construct a reference genome, JG1, by integrating de novo assemblies of three Japanese individuals. After merging the three haploid assemblies constructed using a hybrid scaffolding strategy, we define major variants among the three (i.e., majority decision) and adopt them as the reference allele. We also position the scaffolds along chromosomes with the aid of conventional genetic and RH maps. We then assess the extent to which JG1 represents the major variants in the Japanese population in terms of SNVs and SVs. As an example potential application, we also demonstrate the utility of using JG1 as a reference genome in NGS analyses aimed at identifying the causal variants of several rare diseases. We demonstrate that the multiple-genome integration strategy is effective in constructing a population-specific genome.

## Results

**Construction of JG1.** To construct a genome sequence with population reference-quality, the population background of the reference genome should not significantly diverge from the backgrounds of sample individuals in order to reduce unnecessary variant calls that merely reflect the difference in the population background. In the case of our study, the donor should therefore be chosen from the Japanese population originating

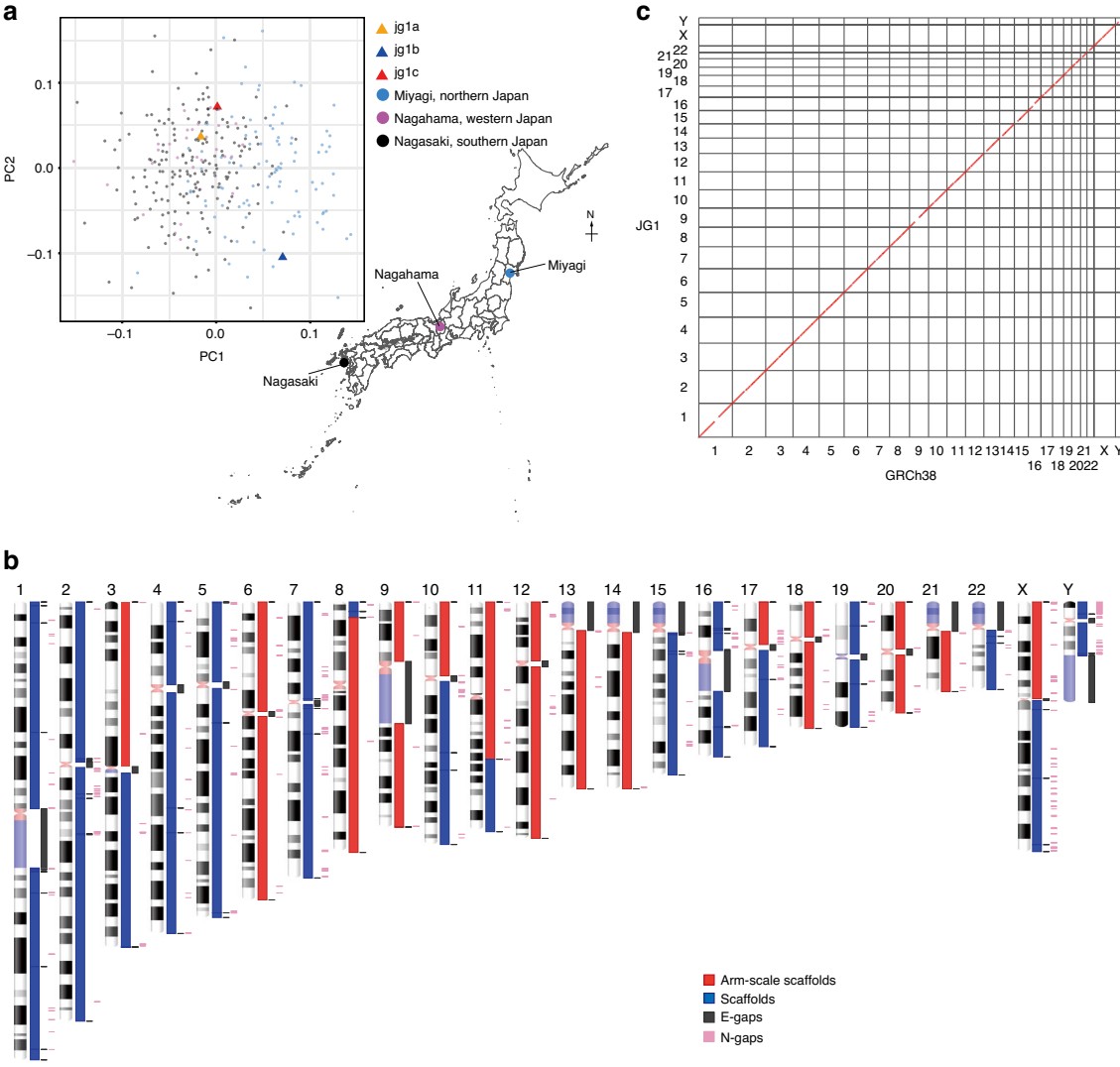

**Fig. 1 Construction of JG1. a** PCA plot showing that the three sample donors are within the Japanese population cluster. The Japan map is retrieved from the Geospatial Information Authority of Japan (https://www.gsi.go.jp/kankyochiri/gm_jpn.html). **b** Idiogram showing the regions sequenced for each chromosome in JG1. Red and blue boxes indicate scaffolds; the red box spans an entire chromosomal arm. Dark gray boxes denote E-gaps, which represent links connected by genetic and RH maps or gaps inserted according to other evidence. Pink boxes denote N-gaps, which are unresolved regions linked by Bionano genome maps, or the putative PAR1 region in the Y chromosome. **c** Harr plot representing the co-linearity between the reference genome GRCh38 and JG1. PCA principal component analysis, RH radiation hybrid, PAR pseudo-autosomal region. Source data for **a** are provided as a Source Data file.

from the main island of Japan. In addition, we built the Japanese reference genome independent of the GRC reference genome in order to eliminate known ethnic biases toward African and European backgrounds as well as any other (and possibly unknown) biases. We therefore performed de novo assembly of the Japanese human genome. Majority-based decision-making regarding multiple de novo assemblies was implemented as an effective way to avoid inclusion of rare reference alleles. This majority-based decision-making strategy produced a haploid genome sequence amenable to analyses using currently available and standard bioinformatics tools for NGS data.

We recruited three male Japanese volunteers, and they were given the sample names jg1a, jg1b, and jg1c (jg1a is the same individual as JPN00001[21]). Principal component analysis (PCA) based on the genotypes inferred by whole-genome sequencing indicated that the subjects were scattered within the cluster of the Japanese population (Fig. 1a). G-banding analyses (Supplementary Fig. 1) indicated that all three individuals had a normal karyotype, although subject jg1a had a common pericentric

inversion within chromosome 9, inv(9)(p12q13). Because it was difficult to assemble the pericentric region of chromosome 9 equally for all three subjects, this variation does not appear to have affected the assembly results (Supplementary Fig. 2).

To construct a reference-quality haploid genome sequence, we integrated the three de novo assembled genomes (see Supplementary Fig. 3 for an overview; see Supplementary Tables 1–3 for materials). For each subject, we sequenced deeply (122× for jg1a, 123× for jg1b, and 128× for jg1c) using PacBio technology (Supplementary Fig. 4 and Supplementary Table 1) and then performed de novo assembly using Falcon software[26]. The de novo assemblies yielded 2194, 2227, and 2120 primary contigs for jg1a, jg1b, and jg1c, respectively (Supplementary Table 4). The contig N50 value was approximately 20 Mb for the three subjects (Supplementary Table 4). Using ArrowGrid software[27], the primary contigs were then error-corrected (polished) with the same long reads used for the initial de novo assembly.

We also obtained deep Bionano optical maps for each subject (123× and 140× for two enzymes for jg1a; 160× and 175× for one

**Table 1 Assembly statistics.**

| | Total length (bp) | Contig[a] | | Scaffold[b] | | Number of misassemblies[c] | N-gap[d] | | Reference |
|---|---|---|---|---|---|---|---|---|---|
| | | number | N50 (Mb) | number | N50 (Mb) | | number | length (Mb) | |
| JG1 | 3,085,782,898 | 1068 | 23.6 | 624 | 142.0 | 1654 | 473 | 251 | this study |
| JG1 metasca ffolds | 2,858,691,982 | 1043 | 20.8 | 708 | 66.4 | 1581 | 338 | 22.5 | this study |
| ZF1 | 2,845,586,846 | 3148 | 23.6 | 2321 | 47.2 | 2066 | 1360 | 7.7 | 25 |
| AK1 | 2,904,207,228 | 3128 | 17.7 | 2832 | 44.8 | 2138 | 264 | 37.3 | 23 |
| HX1 | 2,934,082,568 | 5843 | 8.3 | 5323 | 22.0 | 2688 | 4025 | 39.3 | 24 |
| Swe1 | 3,127,010,000 | 3139 | 9.5 | NA | 49.8 | NA | NA | NA | 20 |
| Swe2 | 3,103,497,000 | 3162 | 8.5 | NA | 45.4 | NA | NA | NA | 20 |

*NA* not applicable.
[a]Contig statistics for JG1 were calculated by splitting JG1 pseudo-molecules at the gap sites.
[b]Scaffold statistics for JG1 were calculated on JG1 pseudo-molecules.
[c]Number of misassemblies were calculated by using Quast-LG software[66] with the reference GRCh38 as the truth set.
[d]Number of gap and gap bases in JG1 includes the heterochromatin regions.

enzyme for jg1b and jg1c, respectively; Supplementary Fig. 5 and Supplementary Table 2) and performed de novo assemblies of these optical maps to generate genome maps (Supplementary Table 5). Each de novo assembly of the Bionano optical maps was performed in two rounds (rough and full) to guarantee independence relative to the GRC reference genome (see Methods section). We then performed hybrid scaffolding between the PacBio-derived contigs and the Bionano-derived genome maps. The resulting hybrid scaffolds were then polished with 55×, 59×, and 57× Illumina short reads for subjects jg1a, jg1b, and jg1c, respectively (Supplementary Table 3). The number and N50 value of the resulting hybrid scaffolds were 1911, 1893, and 1797, and 86.28 Mb, 59.38 Mb, and 58.20 Mb for subjects jg1a, jg1b, and jg1c, respectively (Supplementary Table 4). The number and length of gap regions in the hybrid scaffolds were 417, 413, and 380 and 34.3 Mb, 28.3 Mb, and 24.5 Mb, respectively (Supplementary Table 4; see Supplementary Fig. 6a–c for the gap position). These and other assembly statistics were better than or comparable to other published de novo assemblies (Table 1 and Supplementary Tables 4 and 6). The estimated base-error rate of the three sets of polished hybrid scaffolds was $1.02–1.46 \times 10^{-5}$, being well below the standard base-error rate of $1 \times 10^{-4}$ for reference quality[25] (Supplementary Table 7).

To enhance the quality of our genome assembly, we adopted a meta-assembly strategy in which multiple assemblies were merged to yield a single assembly. In meta-assembly strategies, individual assemblies are aligned, and one best assembly is selected for each aligned segment based on the absence of rare SVs, unresolved sequences, or possible mis-assembly inferred by other experimental evidence, such as mate-pair sequencing data[28]. For meta-assembly, Metassembler software[28] was applied to 37× mate-pair short reads from the three subjects in sum to infer discordance among the individual scaffolds (Supplementary Table 3; see Supplementary Fig. 7 for fragment size distribution). A total of 12 meta-assemblies, or sets of meta-scaffolds, were generated from the three sets of scaffolds, based on the order and combination of the processed sets of scaffolds (see Methods section). Among the 12 possible combinations, we found that one combination (jg1c + (jg1a + jg1b))—which merged the scaffolds of jg1c with the meta-scaffolds generated from those of jg1a and jg1b in this order —exhibited no apparent large chimeric mis-assembly in any autosomes. This combination was chosen for the downstream sophistications; the absence of chimeric mis-assembly was assessed using STS markers described later. This set of meta-scaffolds exhibited better contiguity and accuracy than the original set of scaffolds for subject jg1c (Supplementary Table 4).

Although meta-scaffolds were more contiguous and accurate than individual sets of scaffolds, rare reference alleles should still

have been retained in the meta-scaffolds. To eliminate these rare reference alleles, we aligned the three individual sets of scaffolds against the meta-scaffolds, performed variant calling, defined the major allele among the three sets of scaffolds, and substituted the minor allele on the meta-scaffolds for the major allele in terms of SNVs, indels, and SVs (Supplementary Fig. 8a). For tri-allelic sites, we chose one allele randomly among the three as a reference allele. The total number of multi-allelic sites was 120,937, among which it was estimated that erroneous alleles were chosen at only 30 sites via this random choice (see Supplementary Note 1 for more details). We also found that two assemblies among the three contained a 2.6 Mb inversion in the long arm of chromosome 9 (Supplementary Fig. 2), and we confirmed that the meta-scaffolds also contained the inversion.

We next tried to anchor the majority-voted meta-scaffolds on each chromosome. To do so, we utilized a total of 85,386 distinct STS markers from three genetic maps and six RH maps pre-dated the reference genome: the Genethon[29], deCODE[30], and Marsh-field[31] genetic maps and the Whitehead-RH[32], GeneMap99-GB4[33], GeneMap99-G3[33], Stanford-G3[34], NCBI_RH[35], and TNG[36] RH maps. We searched for in silico amplification of STS markers by electronic PCR analysis of the meta-scaffolds and used ALLMAPS software[37] to order and orient the meta-scaffolds to build chromosomes. The co-linearities between the anchored meta-scaffolds and genetic and RH maps were $0.999 \pm 0.004$ and $0.986 \pm 0.021$, respectively (Pearson's correlation coefficient; mean ± SD). However, we found that ALLMAPS using all nine abovementioned maps did not assign any meta-scaffolds to the Y chromosome, probably because most of the maps did not include the Y chromosome. Nonetheless, we found that ALLMAPS using three of the nine maps (deCODE, TNG, and Stanford-G3) assigned some meta-scaffolds to the Y chromosome as well as autosomes and the X chromosome. Therefore, we adopted the ALLMAPS assignment with the nine maps for autosomal assignment and those with three maps to the sex chromosomes.

After anchoring these meta-scaffolds to chromosomes, we found a chimera in the sex chromosomes. A meta-scaffold harboring the *SRY* locus, a gene on the Y chromosome, was chimeric and anchored to the long arm of the X chromosome in the selected set of meta-scaffolds. We therefore chose a set of meta-scaffolds from another set of meta-scaffolds (jg1a + (jg1b + jg1c)) for the long arm of the X chromosome that had no apparent chimeric region.

We also manually modified the length of unresolved regions in the telomeric, centromeric, and constitutive heterochromatic regions represented as a stretch of Ns (see Supplementary Methods and Supplementary Table 8). We then masked a pseudo-autosomal region (PAR) in the Y chromosome to

guarantee that the resulting sets of sequences represented a haploid. In addition, we shifted the start position of the mitochondrial meta-scaffold to match the revised Cambridge Reference Sequence (rCRS) coordinates[38], which provides the reference coordinate system for the mitochondrial genome.

The procedure described above yielded a set of chromosome-level sequences for 22 autosomes, 2 sex chromosomes, and 1 mitochondrial chromosome, along with 599 unplaced scaffolds, and we designated this set of sequences JG1 (Fig. 1b). The total length of JG1 was approximately 3.1 Gb, including 473 gap regions of 251 Mb in total length, of which 227 Mb was intentionally inserted to represent telomeric, centromeric, and heterochromatic regions (Table 1, Fig. 1b, Supplementary Fig. 6d, and Supplementary Table 8). Notably, in the JG1 genome assembly, 19 chromosomal arms were successfully represented as single scaffolds (Fig. 1b). After constructing these chromosome-level sequences, we then aligned them to reference genome GRCh38 using minimap2 software[39] and found an overall high similarity between the two genomes at the sequence level (Fig. 1c). We also quantified consensus quality using dnadiff software[40] and found that JG1 covered 95.53% of the reference with 99.79% average identity (Supplementary Table 9). Because JG1 was built independently from the reference genome GRCh38, this overall high similarity provided strong support for our approach for building JG1 described above.

To further assess the assembly quality of JG1, we counted the number of protein-truncating variants (PTVs) and found that JG1 harbored only 374 protein-truncating SNVs and 407 such indels, even though JG1 covered 95.53% of the reference GRCh38 (Supplementary Table 9). The total number of PTVs of JG1 was the lowest among other high-quality assemblies. Moreover, we assessed whether JG1 can fill the remaining gaps in the reference GRCh38[24], and found that JG1 uniquely filled 36 gaps, which was the second-highest number of uniquely filled gaps among other assemblies (Supplementary Table 9). The 36 gap-filling sequences in JG1 did not apparently have genic regions (Supplementary Note 2; see also Supplementary Fig. 9 and Supplementary Table 10).

**Representativeness of the JG1 haplotype in terms of SNVs.** To assess whether JG1 is a representative reflection of the SNV composition of the Japanese population, we performed PCA with several settings using JG1, the reference GRCh38, 13 assemblies from diverse populations, and haplotypes constructed from 11 HapMap3 populations (see Methods section and Supplementary Tables 6 and 9 for the derived population of the assemblies). First, we performed PCA with JG1 and 172 Japanese (JPT) haplotypes and confirmed that JG1 was plotted within the Japanese cluster (Fig. 2a; see Supplementary Fig. 10 for plots with PC3). Second, we performed PCA with JG1, 5 other Asian assemblies, and 506 Asian haplotypes constructed from three Asian populations: JPT, Han Chinese (CHB), and Chinese in Denver (CHD) (Fig. 2b). The PCA plot included two distinct clusters (namely, Japanese and others), with the JG1 haplotype associated with the Japanese cluster. Third, we performed PCA with JG1, GRCh38, and world-wide populations, and found that the JG1 haplotype localized near the cluster of Asian populations, whereas the GRCh38 haplotype localized between the African and European populations, as expected, based on the donors' ancestries (Supplementary Fig. 11a). Notably, the JG1 haplotype localized near but outside of the Asian cluster; it localized to the most distant site both from the European and African populations than any other Asian haplotypes, suggesting an "Asianness" when compared with the other two populations. We reasoned that this occurred because JG1 harbors the major allele among the Japanese population in most SNP sites due to the majority decision procedure that removes the minor allele among the Japanese populations. The removed minor allele among the Japanese population can be the major in European or African populations. That would be why JG1 was plotted most distant from the two populations than any Asian haplotype (for further discussion, see Supplementary Note 3, Supplementary Figs. 11 and 12).

To assess whether JG1 harbors the major allele among the Japanese population across SNV sites, we aligned JG1 against the reference genome hs37d5, detected SNVs, and investigated their allele frequency (AF) using the AF panel of 3552 Japanese individuals (namely, the 3.5KJPNv2 AF panel[41]). We chose hs37d5 as the reference because the 3.5KJPNv2 AF panel was built on that reference genome; hs37d5 is a reference genome based on GRCh37/hg19 primary sequences amended mainly with those of other high-quality assemblies, and it is thus well-suited and widely used for NGS analyses[42]. The genome-by-genome alignment and comparison using minimap2 and paftools software[39] called 2,501,575 SNVs between hs37d5 and JG1 in the autosomes and X chromosome. Of these SNVs, 93.9–95.9% were

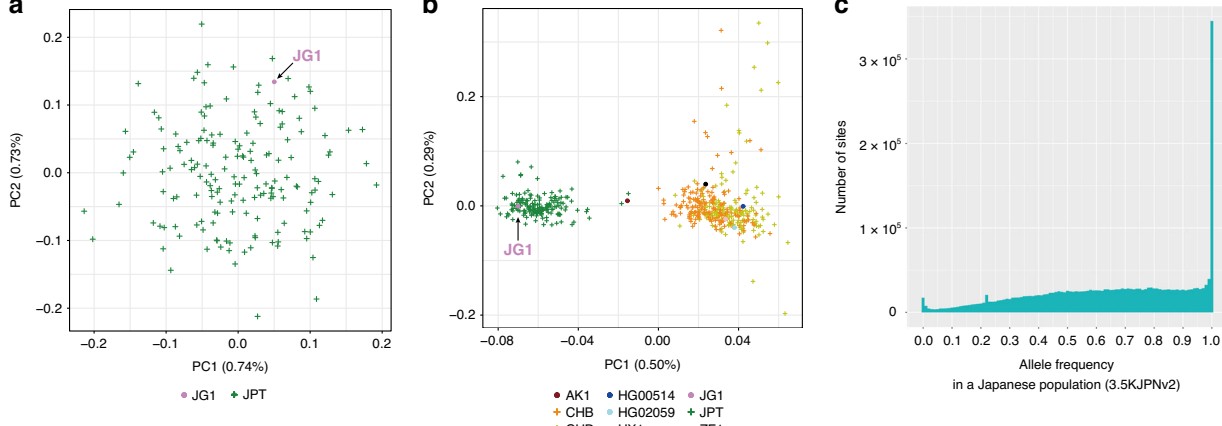

**Fig. 2 SNV characteristics of JG1. a** PCA plot of the haplotype SNP composition of JG1 and HapMap3 Japanese in Tokyo (JPT) samples. **b** PCA plot of the haplotype SNP composition of JG1, the other five Asian assemblies, and Asian samples from HapMap3. The percentages in the X- and Y-axes in **a** and **b** indicate the variance explained by the principal component. Populations for the assemblies are provided in Supplementary Tables 6 and 9. **c** Unfolded site frequency spectrum representing the frequencies of alleles employed in the JG1 sequence in the Japanese population of 3.5KJPNv2. SNV single nucleotide variant. Source data for **a** and **b** are provided as a Source Data file.

validated by two independent mapping analyses, namely MGI DNBSEQ short read and Oxford Nanopore Technologies (ONT) long-read datasets not used for constructing JG1 (see Supplementary Tables 11–13 and Methods section). We then extracted the frequency of the allele employed in JG1 from the 3.5KJPNv2 AF panel to create a site frequency spectrum, in which the horizontal axis indicates the non–hs37d5-type allele and the vertical axis indicates the number of such SNV sites (Fig. 2c). From these data, we found 241,500 SNV sites with an AF = 1.0, indicating that all of the Japanese chromosomes in the AF panel carried the JG1-type allele at the 241,500 sites. This corresponds to 97.99% of all such SNV sites that had an AF = 1.0 (246,464) in the 3.5KJPNv2 AF panel. Similarly, we identified 367,271 and 626,254 SNV sites with an AF ≥ 0.99 or ≥ 0.90, respectively, corresponding to 97.11% and 96.24% of such SNV sites in the 3.5KJPNv2 AF panel, respectively (378,211 and 650,718). A peak observed at an AF of ~0.22 was associated with the SNPs clustered in the XTR region—a region known to harbor complex duplications—within 88.8 to 92.4 Mb on the X chromosome. A peak at an AF of approximately zero could most likely be attributed to artificial SNVs called at the edges of alignments. We also assessed the effectiveness of the majority decision approach. Of the 2,501,575 SNVs, 1,176,922 (47%) and 1,204,762 (48%) were detected in three and two of the three JG1-donor individuals, respectively (Supplementary Fig. 8b).

**Representativeness of the JG1 haplotype in terms of SVs**. To investigate differences between JG1 and GRCh38 in terms of SVs, we aligned JG1 against the reference GRCh38 and detected SVs (insertions and deletions) using the minimap2 and paftools software programs[39]. A genome-by-genome comparison detected 8697 insertions and 6190 deletions > 50 bp in length. The largest insertion and deletion were 15,621 bp and 17,221 bp, respectively; 8 insertions ≥ 10 kb were less reliable because mapping-based orthogonal validation did not detect ones in that range (see below). The length distribution of the SVs exhibited two peaks, at approximately 300 bp and 6 kb (Fig. 3a). We confirmed that the 300-bp and 6-kb peaks were associated with *Alu* and LINE1, respectively. Most of the SV-associated *Alu* and LINE1 were classified as *Alu*Y and L1HS, respectively, both of which constitute the currently active subclass of these transposable elements (the length distributions of detected transposable elements are shown in Supplementary Fig. 13). In addition, the detected SVs were often observed in the sub-telomere–telomere regions (Fig. 3b), consistent with a previous report[12].

To validate the detected SVs, we performed two orthogonal SV analyses based on mapping the PacBio long reads used for constructing JG1 and ONT long reads not used for constructing JG1 (Supplementary Tables 13–15). We mapped PacBio and ONT long reads using NGMLR and minimap2, respectively, and detected SVs using Sniffles software[43]. We found that both mapping-based analyses detected a comparable or larger number of SVs of similar size and chromosomal distribution as well as other types of SVs (Supplementary Tables 14 and 15 and Supplementary Fig. 14), supporting 74.8–90.2% of the detected insertions and 81.3–86.9% of deletions (Supplementary Table 16).

We also compared the detected SVs in JG1 with the other three Asian assemblies: AK1, HX1, and ZF1. Genome-by-genome alignment against GRCh38 detected comparable numbers of both insertions and deletions among the four assemblies, with the exception of slightly lower number in HX1, probably due to its fragmented assembly (Supplementary Fig. 15a). We also performed genome-by-genome alignment-based SV detection using ZF1 as a reference (Supplementary Fig. 15b) and found that fewer insertions and deletions were detected in JG1 compared

with using GRCh38 as the reference (4718 vs 8697 insertions and 5341 vs 6190 deletions against ZF1 vs GRCh38, respectively), probably due to their ancestral closeness and/or similarity in assembly strategies.

To investigate the extent to which JG1 represents a Japanese population in terms of SVs, we mapped short reads of 200 Japanese individuals to JG1 and GRCh38 to compare the average read depth among the 200 individuals around the SVs in JG1 and GRCh38. As shown in Fig. 3c, copy-number gains, representing some of the detected insertions, were typically associated with a 'piling-up' of the average depth, whereas deletions were typically associated with a depression of the average depth in GRCh38. Neither pattern was clearly evident in the corresponding region in JG1 (Fig. 3c), suggesting that most of the Japanese samples shared the SVs. To determine whether this pattern is common among SVs throughout the genome, we compared the maximum difference in average depth between the SV region and its adjacent upstream region of the same length and found that the difference in the average depth was smaller in JG1 than GRCh38 (Fig. 3d; $P = 8.8 \times 10^{-11}$ for $n = 3953$ pairs of insertions; $P < 2.2 \times 10^{-16}$ for $n = 2767$ pairs of deletions; Wilcoxon two-sided signed rank tests).

The reference genome might lack some population-specific sequences[13,14], thus might make some short reads from samples in the population unmapped. To determine whether JG1 has Japanese population-specific sequences not present in the reference genome, we collected the unmapped reads of 1070 Japanese individuals[44] when mapped to the reference GRCh38 and re-mapped them to JG1. Of the $581 \pm 21 \times 10^6$ reads per individual, $4.5 \pm 1.5 \times 10^6$ reads were flagged as unmapped to GRCh38 ($n = 1070$; mean ± SD). Of these, we found that $98,670 \pm 11,798$ reads could be successfully mapped to JG1 with high mapping quality (MAPQ ≥ 20). These reads were mapped to 449,549 distinct regions in JG1, and 164 regions had at least one mapped read from every individual (Supplementary Data 1). Among the 164 regions, 128 exhibited similarity to previously reported non-reference sequences[13,14]. Among the other 36 regions, 34 exhibited 89–100% identity to a previously reported human genome sequences with matched chromosomal origin if known (26/34 regions; Supplementary Data 2)[45]. One of the remaining two regions was 3 kb in length and exhibited 88.7% identity for approximately only half of the region (1469 bp) with a glucoside xylosyltransferase 1 from *Pan paniscus* (XM_014347211.2). The other region, which was 1.5 kb in length and found in chromosome 7, exhibited only 85.5% identity to a chimpanzee BAC clone CH251-285G22 sequence (AC190217.3) from a matched chromosome.

**Utility of JG1 as a reference for exome analyses**. To evaluate whether JG1 is a suitable reference genome for clinical NGS analyses, we examined exomes of Japanese families with rare diseases[46]. The sample cohort consisted of 22 individuals from six trio families and one quartet family. All of the families had one child affected with diplegia, and eight causal variants were identified in previous analyses using the reference genome hg19 (Table 2). The diseases exhibited autosomal recessive, compound heterozygous, and autosomal dominant modes of inheritance, including de novo mutations, and the causal variants included both single nucleotide and deletion variants.

To facilitate exome re-analysis with JG1, we lifted over the resource bundles of the Genome Analysis ToolKit (GATK) software (which are used for accurate variant calling) and GENCODE gene annotation information[47] to predict variant effects (see Methods section) and performed exome analyses according to GATK best practices. For comparison, we also

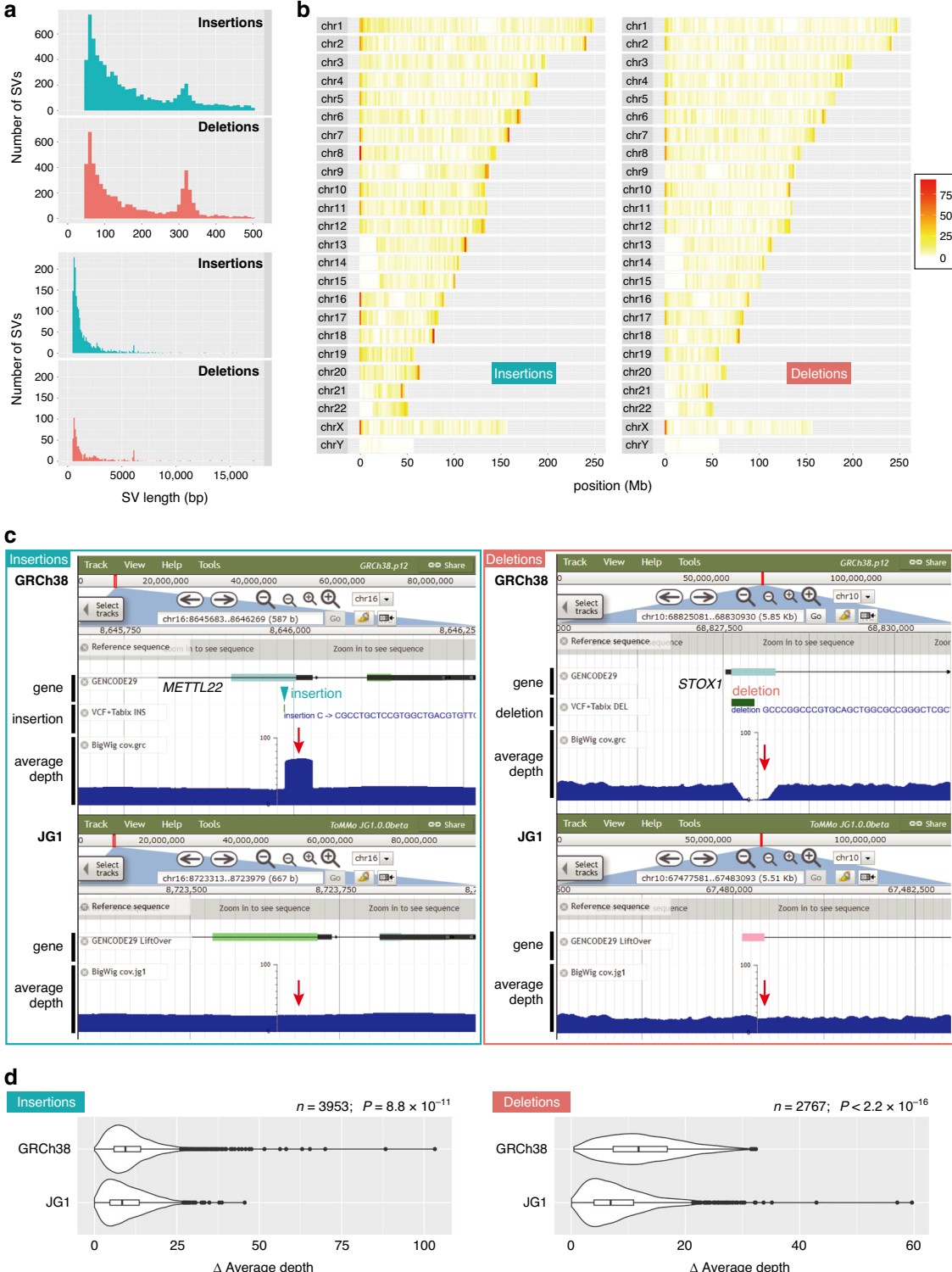

**Fig. 3 Analysis of JG1 SV. a** Length histogram of small (≤ 500 bp) and large (> 500 bp) insertions and deletions detected by comparing JG1 and GRCh38. **b** Distribution of insertions and deletions among the chromosomes of GRCh38. **c** JBrowse snapshots of one insertion/deletion example. Tracks are GENCODE gene annotations, detected SVs, and average depth of short reads from 200 Japanese samples. **d** Difference in average depth between the SV and upstream regions of same length as the SV for GRCh38 and JG1. Violin plots and boxplots are shown ($n = 3953$ and 2767 pairs of insertions and deletions, respectively; bold bars indicate the median, boxes indicate 25th–75th percentile regions, and whiskers indicate 1.5× interquartile ranges with outliers). $P = 8.8 \times 10^{-11}$ for $n = 3953$ pairs of insertions; $P < 2.2 \times 10^{-16}$ for $n = 2767$ pairs of deletions; Wilcoxon two-sided signed rank tests. SV structural variant. Source data for **d** are provided as a Source Data file.

**Table 2 Diplegia cohort variant characteristics.**

| IDᵃ | Family | Type | Locus | Variant(s) | Variant effect prediction | Mode of inheritance |
|---|---|---|---|---|---|---|
| 3 | FD-05 | trio | CTNNB1 | c.1683 + 2 T > C | HIGH (splice donor) | de novo SNV |
| 5 | FD-07 | trio | CYP2U1 | c.651delC | HIGH (frame shift) | autosomal recessive deletion |
| 6 | FD-08 | trio | SPAST | c.1276 C > T | MODERATE (missense) | de novo SNV |
| 7 | FD-09 | quartet | GNAO1 | c.736 G > A | MODERATE (missense) | de novo SNV |
| 9 | FD-11 | trio | CACNA1A | c.653 C > T | MODERATE (missense) | de novo SNV |
| 10 | FD-12 | trio | SPAST | c.1496 G > A | MODERATE (missense) | de novo SNV |
| 11 | FD-13 | trio | AMPD2 | c.515 + 1 G > A; c.1724C > T | HIGH (splice donor); MODERATE (missense) | compound heterozygous SNV |

ᵃCase ID from Table 2 in Takezawa et al.[46].

performed exome analyses using hs37d5 as the reference; we used hs37d5 because the exome capture sequence was designed using the GRCh37/hg19 sequence. The JG1-based exome analyses correctly identified all (8/8) of the previously reported causal variants. In addition, the total number of called variants was lower in JG1 than the reference hs37d5 (Fig. 4a). This comparison was done in the 225,888 exome regions with one-to-one correspondence between JG1 and hs37d5 (87,971,409 bp and 87,997,786 bp for JG1 and hs37d5, respectively). Moreover, the number of both high- and moderate-impact variants (which are the primary causal variant candidates) was also lower in JG1 than hs37d5 (Fig. 4b; 473 ± 16 vs 671 ± 13 high-impact and 8774 ± 97 vs 10,599 ± 89 moderate-impact variants for JG1 and hs37d5, respectively; mean ± SD). These findings suggest that JG1 produces fewer total candidate variants while successfully detecting disease-causing variants in whole-exome analyses.

To validate the variants detected via the exome analysis using JG1 as a reference, we Sanger-sequenced the 8 regions harboring the disease-causing variants of the 22 individuals from 7 families (Supplementary Fig. 16), along with an additional 42 regions of 44 individuals from 24 families. The total Sanger-sequenced length was 25,896 bp, with mean Phred score of 33.9. The sequenced regions corresponded to 50 distinct regions 12,081 bp and 12,088 bp in length for hs37d5 and JG1, respectively. Within these 50 regions, we detected 58 variant calls from both exome analyses, and 57 calls were ascertained by Sanger sequencing for both JG1 and hs37d5, corresponding to a positive predictive value of 98.3% (Supplementary Table 17). The sensitivity was also the same between these two genomes (57/59; 96.6%; Supplementary Table 17). These results suggest that variant calls from JG1 are as accurate as those from the reference hs37d5.

In addition, we compared the variants detected with JG1 and hs37d5 by lifting over the JG1-detected variants to hs37d5 and found ~15,000, ~29,000, and ~51,000 specific to JG1, hs37d5, and both references, respectively (Fig. 4c). Moreover, we extracted the non–GRC-type AF in the JG1-specific, hs37d5-specific, and shared variant sites from the 3.5KJPNv2 AF panel and found that most of the hs37d5-specific variants were major alleles among the Japanese population, whereas the shared and JG1-specific variants tended to be biased toward the minor AFs (Fig. 4d). We also assessed whether JG1 can serve as the reference for whole genome sequencing (WGS) analysis by mapping WGS short reads of 1070 Japanese individuals to JG1 and comparing the non–GRC-type AF with those of the reference hs37d5. We found that the non–GRC-type AF was almost equivalent between the two references throughout the wide allele frequency range ($R > 0.999$) for the 19,772,783 SNV sites called in autosomes of both references (Supplementary Fig. 17).

## Discussion

Here, we report the construction of a Japanese haploid genome sequence, JG1, by integrating three highly contiguous de novo

hybrid assemblies from three Japanese donor individuals to build a population-specific (i.e., ethnicity-matched) reference genome. Employing a meta-assembly approach produced a more contiguous and accurate assembly, and relying on majority decision among the three genomes substituted most of the rare reference alleles. The results of both SNV and SV analyses suggested that the JG1 haplotype represents major variation among the Japanese population. Moreover, we demonstrated that JG1 exhibits an advantage as an ethnicity-matched reference, at least for NGS analyses within the clinical context of whole exomes of Japanese samples. Using JG1 could thus facilitate detecting the proverbial needle in a haystack, by reducing the size of the haystack in NGS analyses of the Japanese population.

The accuracy of JG1 was comparable or superior to other high-quality genomes especially in terms of the number of PTVs (Supplementary Table 9). The remaining PTVs, however, might be removed by a genic region-specific indel error-correction pipeline[48]. Although application of the pipeline to our data reduced PTVs in the individual assemblies (Supplementary Table 18), careful examination of the pipeline revealed that it requires information from the reference GRCh38/hg38. Since our purpose was to construct a genome assembly completely independent of the reference genome, we did not adopt this pipeline. Nonetheless, because our majority decision strategy among the three assemblies was as effective as the pipeline, the current version of JG1 deserves the reference quality assembly.

Integration and majority decision regarding multiple assemblies to yield a single haploid genome can produce a highly contiguous and accurate assembly, thus effectively eliminating most rare reference variations. Haploid representation of the genome is beneficial because it is compatible with many conventional bioinformatics tools developed to date for mapping, variant calling, predicting variant effects, and subsequent interpretations. Although we appreciate that the development of a pan-human genome graph could be the next milestone reached in comprehensively assessing human genetic variations among diverse populations, we expect that population-specific reference genome such as JG1 will prove to be practical and beneficial options for genome analyses of individuals originating from the population.

Several limitations of the current version of JG1 should be noted: (1) sequence incompleteness and gaps/un-localized fragments remain, which could result in erroneous mapping and variant calling; (2) few original annotations on the JG1 coordinates; and (3) incomplete representation of the major variations in the Japanese population. The incompleteness of the genome sequence could be largely overcome by applying other genome sequencing technologies, including ONT ultra-long reads in combination with targeted cloning from whole-genome BAC libraries. Chromosome-scale scaffolding using Hi-C[49] could also contribute to the generation of more contiguous assemblies. The genome of a Japanese complete hydatidiform mole, characterized

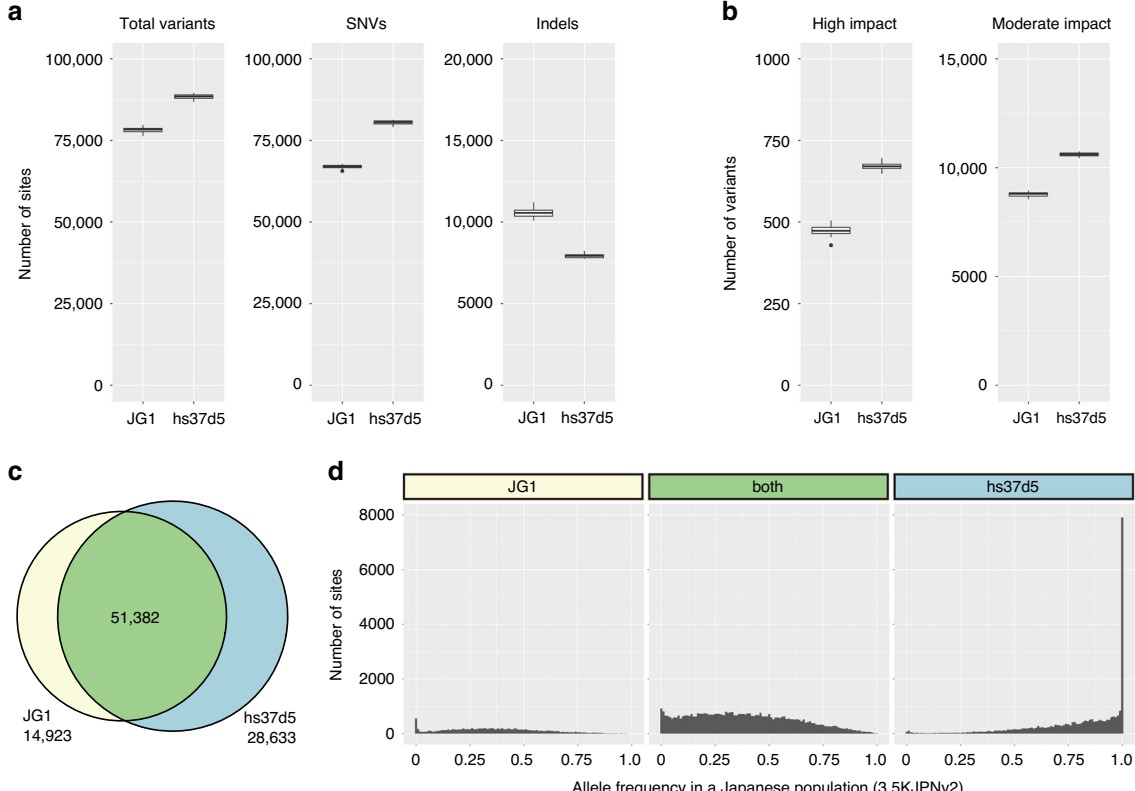

**Fig. 4 Comparison of variants called in exome analyses employing JG1 or hs37d5 as a reference genome. a** Number of total variants, SNVs, and short indels called per individual. **b** Number of high- and moderate-impact variants. Boxplots are shown ($n = 22$ individuals; bold bars indicate the median, boxes indicate 25th–75th percentile regions, and whiskers indicate the ranges without outliers or 1.5× interquartile ranges with outliers). **c** Venn diagram showing overlap relationships between variants detected in JG1 (lifted over to the hs37d5 coordinates) and those detected in hs37d5. Shown are results for a representative individual. **d** Unfolded site frequency spectra representing the frequency of non–GRC-type alleles in the variant sites detected specifically in JG1, in both genomes, and specifically in hs37d5, respectively. Shown are results for the same individual as in **c**. SNV single nucleotide variant. Source data for **a** and **b** are provided as a Source Data file.

by a duplicated haploid genome, could also contribute to gap-filling due to ease of assembly[50]. The limitation of few original annotations could be overcome by constructing an AF panel with JG1 as the reference and by more thorough de novo predictions or experimental inference regarding gene regions. More comprehensive lifting-over of many annotations would also be practically important. The representativeness of the major alleles would be improved by adding more assemblies. One approach that could be used for addition is the phased diploid assembly[26], which provides a pair of haplotype (i.e., diplotype) assemblies from a single individual. Because the two haploid genomes can be regarded as a random sample from a panmictic population, assembling two haploid genomes per individual can increase the representativeness of variations—especially structural variations—in the population. Despite its limitations described above, the current version of JG1 represents a useful tool for efficient causal variant detection.

Additional samples, for example hundreds of samples from a single population, would be beneficial for constructing population-specific reference genomes in the future, not only with respect to SNVs but also SVs, although less is known regarding the entire repertoire of SVs present in a population than that of SNVs. Both integrative haploid reference genomes such as JG1 and collective genome references developed in the future such as genome graphs—both of which can be constructed from hundreds of de novo assemblies—should advance the accuracy of genome analyses and facilitate development of personalized medicine approaches.

## Methods

**Ethics declarations**. This study was approved by the Research Ethical Committee of Tohoku Medical Megabank Organization and the Ethics Committee of Tohoku University Graduate School of Medicine.

### Selection and analysis of donor individuals

*Donor selection.* Three adult male Japanese volunteers were recruited (45–59 years old) and participated in this study with written, informed consent. They were self-reportedly healthy without genetic diseases and had Japanese ancestry.

*G-banding analysis (Supplementary Fig. 1a–c).* G-banding analyses for the three volunteers were performed using phytohemagglutinin-stimulated lymphocytes at the laboratory of SRL Inc. (Tokyo, Japan).

*PCA of donors with Japanese samples (Fig. 1a).* Paired-end reads with length of 162 bp from the three donors (jg1a, jg1b, and jg1c) were individually mapped to hs37d5.fa, and variant calling was performed following GATK best practices[41]. The resulting VCF file was subjected to PCA using EIGENSOFT software (ver. 4.2). We chose 310 Japanese samples from the 3.5KJPNv2 cohort[41]; 100 from Miyagi Prefecture in northern Japan; 29 from Nagahama City, in western Japan; and 181 from Nagasaki Prefecture, in southern Japan. Variants shared among the 313 samples were selected and filtered using plink software (ver. 1.9) with the '--geno 0.05 --maf 0.05 --hwe 0.05', and '--indep-pairwise 1500 150 0.03' options. The resulting dataset consisted of 18,658 variants.

### Genome analyses

*Long-read SMRT sequencing.* Long-read SMRT sequencing was performed as below. Briefly, genomic DNA from nucleated blood cells was sheared to ~20 kb and used for library preparation with a DNA template prep kit 2.0 (Pacific Biosciences; Menlo Park, CA). Size selection was carried out using the Blue Pippin system (Sage Science; Beverly, MA), targeting 18 kb (10–15 kb for some libraries of jg1a). The libraries were sequenced on a PacBio RSII instrument using P6-C4 chemistry. The long-read dataset of subject jg1a was previously described[21].

*Optical mapping.* Optical mapping was performed using the Irys system or Saphyr system, according to the manufacturer's protocol (Bionano Genomics; San Diego, CA). For sample jg1a, high-molecular-weight genomic DNA from nucleated blood cells was nicked using the endonucleases Nt.BspQI or Nb.BssSI and then labeled with fluorophore-tagged nucleotides. The labeled DNA was imaged on the Irys system. For samples jg1b and jg1c, high-molecular-weight genomic DNA from nucleated blood cells was labeled using direct labeling and staining (DLS) chemistry. The labeled DNA was imaged on the Saphyr system.

*Short-read paired-end sequencing.* Short-read paired-end sequencing on a HiSeq 2500 system (Illumina; San Diego CA) was performed as below. Briefly, genomic DNA from buffy coat samples was fragmented to an average target size of 550 bp, and then subjected to library construction using a TruSeq DNA PCR-Free HT sample prep kit (Illumina). The libraries were sequenced on a HiSeq 2500 system with a TruSeq Rapid PE Cluster kit (Illumina) and TruSeq Rapid SBS kit (Illumina) to obtain 162- or 259-bp paired-end reads.

Short-read paired-end sequencing on a DNBSEQ-G400RS system (MGI; Shenzhen, China) was performed as described below. Fragmented genomic DNA targeted to an average size of 400 bp was used for library preparation using PCR-Free DNA Library Prep set V1.0 (MGI), as directed by the manufacturer. After quantification, the libraries were sequenced on a DNBSEQ-G400RS (MGI) system with a DNBSEQ-G400RS High-throughput Sequencing Set V1.0 (MGI) to obtain 150-bp paired-end reads.

*Mate-pair sequencing.* Genomic DNA from nucleated blood cells was used for library construction with a Nextera Mate Pair Library Preparation kit (Illumina), according to the manufacturer's gel-free protocol, which produces a broader range of fragment sizes (2–15 kb). The obtained libraries were size-selected to 300–800 bp (peak at 500 bp) using AMPure XP beads (Beckman Coulter; Indianapolis, IN) and sequenced on a HiSeq 2500 system (Illumina) with a TruSeq Rapid PE Cluster kit (Illumina), and TruSeq Rapid SBS kit (Illumina) to obtain 201-bp paired-end reads.

*Long-read nanopore sequencing.* Genomic DNA extracted from whole blood using Gentra Puregene Blood kit (Qiagen; Hilden Germany) was sheared using a Covaris g-tube device or 26-gauge needle to prepare long or super-long reads, respectively. The sequencing libraries were prepared using an SQK-LSK109 ligation kit (Oxford Nanopore Technologies; Oxford, UK). Sequencing was performed on MinION devices with R9.4.1 flow cells (ONT). The squiggle data obtained from MinION sequencers were subjected to base-calling using Guppy software (version 3.2.4). Reads with Phred-scaled quality > 6 were used after cropping their head and tail 100 bp for mapping.

**Overview of the computational methods for JG1 construction**. A diagram showing an overview of the construction of JG1 is provided in Supplementary Fig. 3. JG1 was constructed according to the following steps, which are also described in the download page for the JG1 sequence file from the jMorp website (https://jmorp.megabank.tohoku.ac.jp/dj1/datasets/tommo-jg1.0.0.20190424/files/tech-notes-for-computation.pdf). The computation was carried out by using the Tohoku Medical Megabank Organization (ToMMo) Super Computer (https://sc.megabank.tohoku.ac.jp/en/outline).

*De novo assembly of PacBio subreads.* PacBio subreads were assembled using Falcon software[26] (build ver. falcon-2017.11.02-16.04-py2.7-ucs2.tar.gz) with the following configurations: reads shorter than 9 kb were used for error-correction of the longer reads ('length_cutoff = 9000'), and error-corrected reads longer than 15 kb were used for assembly ('length_cutoff_pr = 15000'). Detailed settings are provided below:

length_cutoff = 9000 length_cutoff_pr = 15000 genome_size = 3200000000
pa_HPCdaligner_option = -v -dal128 -t16 -e.75 -M16 -l4800 -k18 -h480 -w8 -s100 -T1
ovlp_HPCdaligner_option = -v -dal128 -M24 -k24 -h1024 -e.96 -l2500 -s100 -T1
pa_DBsplit_option = -x500 -s400 ovlp_DBsplit_option = -s400
falcon_sense_option = --ouput_multi --min_idt 0.70 --min_cov = 4 --max_n_read 200 --n_core 1 overlap_filtering_setting = --max_diff 60 --max_cov 60 --min_cov 0 --n_core 12

The contigs were then polished with the PacBio subreads using ArrowGrid software[27] (ver. 81b03f1; GitHub commit tag), with slight modifications to accommodate our number of data files and UGE settings.

*De novo assembly of Bionano optical maps.* We obtained two sets of Bionano optical maps using two different enzymes, Nt.BspQI and Nb.BssSI, for subject jg1a, and one set of Bionano optical maps was obtained with DLE-1 for jg1b and jg1c. In both cases, the Bionano optical maps were assembled in two steps—a rough assembly step and a full assembly step—to perform de novo assembly as independently as possible from the reference. For the rough assembly step for jg1a, we ran pipelineCL.py software using the following settings:

-T 128 -j 8 -f 0.2 -i 0 -b ${data}/Molecules.bnx -l ${work} -V 0 -A -z -u -m
-t ${bin}/Solve3.1_08232017/RefAligner/6700.6920rel/avx/ -a ${bin}/Solve3.1_08232017/RefAligner/6700.6920rel/optArguments_nonhaplotype_irys.xml -C ${work}/clusterArguments_${ver}.xml

For the full assembly step for subject jg1a, we ran the software using the following settings:

-T 128 -j 8 -f 0.2 -i 5 -b ${data}/Molecules.bnx -l ${work} -V 0 -y -m
-t ${bin}/Solve3.1_08232017/RefAligner/6700.6920rel/avx/ -a ${bin}/Solve3.1_08232017/RefAligner/6700.6920rel/optArguments_nonhaplotype_irys.xml -C ${work}/clusterArguments_${ver}.xml -r ${rough_assembly_output}/exp_mrg0/EXP_MRG0A.cmap

For the rough assembly step for subjects jg1b and jg1c, we ran the software using the following settings:

-f 0 -i 5 -b ${data}/all.bnx -l ${work} -V 0 -N 4 -R
-t ${bin}/Solve3.2.1_04122018/RefAligner/7437.7523rel/avx/ -a ${bin}/Solve3.2.1_04122018/RefAligner/7437.7523rel/avx/optArguments_nonhaplotype_DLE1_saphyr_human.xml
-C ${work}/clusterArgumentsBG_saphyr_phi_${ver}.xml

For the full assembly step of subjects jg1b and jg1c, we ran the software using the following settings:

-f 0 -i 5 -b ${data}/all.bnx -l ${work} -V 0 -N 4 -R -y
-t ${bin}/Solve3.2.1_04122018/RefAligner/7437.7523rel/avx/ -a ${bin}/Solve3.2.1_04122018/RefAligner/7437.7523rel/avx/optArguments_nonhaplotype_DLE1_saphyr_human.xml
-C ${work}/clusterArgumentsBG_saphyr_phi_${ver}.xml
-r ${rough_assembly_output}/exp_mrg0/EXP_MRG0A.cmap

The '-T' and '-j' options were varied for computational efficiency. The BionanoSolve software suite was used for the above computation. We used BionanoSolve (ver. 3.1) for the assembly for subject jg1a, and ver.3.2 for the assembly for subjects jg1b and jg1c.

*Hybrid scaffolding.* Hybrid scaffolding was performed using BionanoSolve software (ver. 3.2). Hybrid scaffolding for subject jg1a was performed in the two-enzyme hybrid scaffolding mode using the runTGH.R script with the following options:

-N ${jg1a}-p_ctg.arrow.fa -e1 BSPQI -e2 BSSSI
-b1 ${BspQI_work}/contigs/exp_refineFinal1/EXP_REFINEFINAL1.cmap
-b2 ${BssSI_work}/contigs/exp_refineFinal1/EXP_REFINEFINAL1.cmap
-O ${jg1a_hybscf}/${prefix}
-R ${bin}/Solve3.2.1_04122018/RefAligner/7437.7523rel/avx/RefAligner ${bin}/Solve3.2.1_04122018/HybridScaffold/04122018/TGH/hybridScaffold_two_enzymes.xml

Hybrid scaffolding for subjects jg1b and jg1c was performed in the single-enzyme hybrid scaffolding mode, using the hybridScaffold.pl script with the following options:

-n ${arrow_work}/${individual}-p_ctg.arrow.fa
-b ${bionano_work}/contigs/exp_refineFinal1/EXP_REFINEFINAL1.cmap
-c ${hybscf_work}/hybridScaffold_DLE1_config.tmem.xml
-r ${bin}/Solve3.2.1_04122018/RefAligner/7437.7523rel/avx/RefAligner
-o ${work} -B 2 -N 2 -f
-e ${bionano_work}/contigs/auto_noise/autoNoise1.errbin

*Error correction with short reads.* Two sets of Illumina paired-end short reads with lengths of 162 bp and 259 bp were mapped to the hybrid scaffolds using BWA-MEM software[4] (ver. 0.7.17) with the option '-t 22 -K 1000000'. The alignment file was coordinate-sorted and compressed using the Picard tools (ver. 2.18.4) SortSam command. The resulting BAM files for the 162- and 259-bp paired-end reads were merged using the Picard tools MergeSamFiles command. The merged BAM files were then split to each scaffold using SAMtools[51] (ver. 1.8) view command, and then each scaffold was polished using Pilon software[52] (ver. 1.22, modified to correct the issue reported at https://github.com/broadinstitute/pilon/issues/48) with the option '--threads 22 --diploid --changes --vcf --tracks'. The FASTA files for each polished scaffold were then merged into a single multi-FASTA format file.

*Meta-assembly.* The three sets of polished scaffolds were then meta-assembled using Metassembler software[28] (ver. 1.5; with modification of the type of 'total-Bases' variable in the CEstat.hh from int to long to accommodate large genomes). There were 12 possible combinations to meta-assemble the three sets: (a + (b + c)), (a + (c + b)), ((a + b) + c), ((a + c) + b), (b + (a + c)), (b + (c + a)), ((b + a) + c), ((b + c) + a), (c + (a + b)), (c + (b + a)), ((c + a) + b), and ((c + b) + a), where x + y indicates meta-assemble x and y in this order. For each round of meta-assembly, we aligned the two assemblies using the NUCmer command of MUMmer software[53] (ver. 4.0.0beta2) with the option '--maxmatch -c 50 -l 300'. The resulting DELTA file was filtered using delta-filter software with the option '-1' to extract one-to-one correspondence. Next, the DELTA file was converted to COORDS format using the show-coords command with '-clrTH' option. Short mate-pair reads were classified into four categories (mp, pe, se, and unknown) using NxTrim software[54] (ver. 0.4.3), and the resulting set of reads with the correct mate-pair orientation (mp) were mapped using Bowtie2 software[55] (ver. 2.3.4.1) with the '--minins 1000 --maxins 16000 --rf' options. The output SAM file was

then processed using the mateAn command with '-A 2000 -B 15000' option, indicating that the range of insert length was 2–15 kb. The NUCmer alignment information and the mate-pair mapping information were integrated using the asseMerge command with '-i 5 -c 6' option. Finally, the resulting METASSEM file was converted to FASTA format using the meta2fasta command.

*Major allele substitution.* The three sets of polished hybrid scaffolds were aligned to the 12 sets of meta-scaffolds using minimap2 (ver. 2.12), and variants were called using the paftools call command. After normalizing the manner of variant representation using the BCFtools norm command (ver. 1.8), SNVs and SVs shared by two of the three genomes were detected using the BCFtools isec command, and these were regarded as the major alleles and employed in JG1 using the BCFtools consensus command. For multi-allelic sites, one allele was chosen randomly.

*Detection of in silico STS marker amplification.* We detected in silico amplification of the STS markers in the three genetic and six RH maps (Genethon, Marshfield, and deCODE genetic maps; GeneMap-G3, GeneMap99-GB4, TNG, NCBI_RH, Stanford-G3, and Whitehead-RH maps) from the meta-scaffolds using the in-house electronic PCR software gPCR (ver. 2.6a) with the '-S -D' option ('-S' to show amplicon sequence, '-D': to show direction of markers). The STS markers were obtained from the UniSTS database (ftp://ftp.ncbi.nih.gov/pub/ProbeDB/legacy_unists/). The results were used to infer the presence of chimeric scaffolds. One set of meta-scaffolds (jg1c + (jg1a + jg1b)) was selected for the primary downstream analysis. In addition, to build the X and Y chromosomes, an additional set of meta-scaffolds (jg1a + (jg1b + jg1c)) was selected.

*Anchoring scaffolds to chromosomes.* The electronic PCR results were converted to BED format files, and the coordinates of some RH maps were scaled to approximately 2000 to fit those for the genetic maps; this was done to make it easier to understand the visualization results of the ALLMAPS software[37] (ver. 0.8.12) but did not affect the anchoring results. These maps were merged using the ALLMAPS mergebed command, and then processed using the ALLMAPS path command with the option '--gapsize=10000' to anchor the meta-scaffolds to the chromosomes. The weights of each of the three genetic maps were set to 5, and that of each of the RH maps was set to 1 in the weights.txt file. To anchor the sex chromosomes, three maps (deCODE, TNG, and Stanford-G3) that could anchor some scaffolds to the Y chromosome were used.

*Manual modification.* Consecutive N-gap length for heterochromatic regions was manually modified (see Supplementary Methods for details). The length of consecutive Ns for each chromosome is provided in Supplementary Table 8.

*Building the X and Y chromosomes.* We noted that one set of meta-scaffolds, (jg1c + (jg1a + jg1b)), contained a chimeric scaffold between the long arm of the X chromosome and the *SRY* locus of the Y chromosome. To reduce the chimeric meta-scaffolds, we chose apparently non-chimeric scaffolds anchored to the long arm of the X chromosome from another set of meta-scaffolds, (jg1a + (jg1b + jg1c)), and linked them to the scaffold of the short arm of the X chromosome.

*Masking the pseudo-autosomal region.* To locate the pseudo-autosomal regions, we aligned both the X and Y chromosomes from JG1 using minimap2 with the option '-cx asm5', and vice versa. The alignment started from the terminal region of the Y chromosome and ended at 2.26 Mb. This region was regarded as the putative PAR1 region. Other regions such as PAR2 and XTR were probably unresolved for unknown reasons. The putative PAR1 region was masked using the BEDTools software[56] (ver. 2.27.1) maskfasta command.

*Mitochondrial chromosome.* We aligned the set of meta-scaffolds to GRCh38, the mitochondrial sequence of which was obtained from the rCRS using minimap2 with the option '-cx asm5' to identify a scaffold that corresponds to the mitochondrial genome. We found a scaffold of 16,568 bp in length corresponding to the mitochondrial sequence in another set of meta-scaffolds (jg1a + (jg1b + jg1c)). The start site of the scaffold and the rCRS sequence differed; therefore, we shifted the start site of the scaffold to match that of the rCRS sequence.

## Assembly assessment

*Choice of the reference genome.* GRCh37/hg19 and GRCh38/hg38 are the two most widely used reference genomes. For analyses using short-read mapping, we used hs37d5, an optimized GRCh37 for NGS analysis, except for the non-reference sequence analysis. For other analyses, we used an analysis-ready version without ALT contigs of the latest reference genome, GRCh38, which was downloaded from the Illumina iGenome website (ftp://ussd-ftp.illumina.com/Homo_sapiens/NCBI/GRCh38/Homo_sapiens_NCBI_GRCh38.tar.gz).

*Base-error rate estimation.* The 162-bp and 259-bp paired-end reads were mapped to draft assemblies for each individual using BWA-MEM software. After marking

duplicated reads, variants were called using GATK (ver. 4.1.2.0) HaplotypeCaller (normal mode) with the options '--pcr-indel-model NONE'. Variants within the callable region ($5 \leq$ depth $< 150$) and $GQ \geq 60$ were selected. Sites with a homozygous alternative allele (1/1) genotype were regarded as erroneous for each draft assembly. Read depth was analyzed with mosdepth (ver. 0.2.8) software.

*Consensus quality.* Assemblies were aligned to GRCh38 using the NUCmer command of MUMmer software (ver. 4.0.0beta2). Proportion of covered region and average identity were calculated using dnadiff software[40].

*Protein-truncating variants.* Assemblies were aligned to the reference GRCh38 using minimap2 software, and variants were called using the paftools call command. After normalization of indels using the BCFtools norm command, variants were annotated using SnpEff software[57] with the GRCh38.86 database. Variants with HIGH impact (e.g., frameshift variants, splice acceptor/donor variants, stop gain variants, etc.) were counted as protein-truncating variants.

*Gap-filling of the reference genome.* The gap-filling ability of each assembly was estimated using uniline software (ver. 99969cc; GitHub commit tag) gfa.pl script[24].

**Idiogram drawing (Fig. 1b).** Idiograms were depicted using JG1 BED files scaled to 90% of the original length so that the drawing of JG1 chromosomes longer than that of GRCh38 would be successful using the NCBI Genome Decoration Page (https://www.ncbi.nlm.nih.gov/genome/tools/gdp). The lengths of the chromosomal arms and the centromeric regions of the idiograms were manually modified to fit the scaffold length of JG1.

**Possible shared large inversion (Supplementary Fig. 2).** Two large scaffolds corresponding to chromosome 9 were extracted from each assembly using the faSomeRecords command. Orientation was carried out by using the seqtk software (ver. 1.3) 'seq -r' command. Next, the chromosome 9 sequence from GRCh38 and the two large scaffolds from each subject were aligned using minimap2 (ver. 2.12) with the '-t 12 -x asm5 --cs' option. Harr plots were drawn using the minidot command (bundled with miniasm software[58] [ver. 0.2]) with the '-L' option. The idiogram of chromosome 9 was drawn using the NCBI Genome Decoration Page.

**PCA (Fig. 2a, b, and Supplementary Figs. 10–12).** Variants were filtered from 1011 HapMap3 diploid individuals data[59] using plink software with '--make-bed --geno 0.05 --maf 0.05'. The resulting filter-passed 1,205,633 variants and 1011 diploid individuals were subjected to the following analysis. The variants from all of the 13 haploid assemblies were extracted by first aligning each assembly onto the reference hg18, which was used as the coordinate for the HapMap3 variants, and then processed using a modified LiftMap.py script (https://genome.sph.umich.edu/wiki/LiftMap.py). Nine assemblies without chromosomal-scale scaffolding (AK1, HX1, HG00268, HG00733, HG01352, HG02059, HG02106, HG02818, and HG04217) were scaffolded using GRCh38 before alignment using RaGOO software[60] to reduce misalignments and enable selection of autosomal variants. After selecting autosomal variants, removing tri-allelic sites, and merging the 2022 HapMap3 haploids and 13 assemblies, 546,871 variants remained as those shared among the 2035 haploids (for Supplementary Fig. 11a). The 546,871 variants were then filtered and pruned using plink software with the '--make-bed --geno 0.05 --maf 0.05' and '--indep-pairwise 1500 150 0.03' options. The resulting 23,033 variants and 2035 haploids were subjected to PCA using EIGENSOFT software.

For six haploid assemblies and three HapMap3 Asian populations (for Fig. 2b), 506 haplotypes from the JPT, CHB, and CHD populations were chosen. Four CHD samples were omitted due to apparent inconsistency inferred from a pre-analysis of the PCA plots including these samples. The resulting 26,727 variants and 512 haploids were subjected to PCA.

For PCA of JG1 and 172 HapMap3 JPT haplotypes (for Fig. 2a), 9874 pruned variants and 173 haploids were subjected to PCA.

The other PCAs followed the same quality control procedure. For PCA of JG1 and African and Asian populations (Supplementary Fig. 11b), 20,499 pruned variants and 1023 haploids were subjected to PCA. For PCA of JG1 and European and Asian populations (Supplementary Fig. 11c), 27,360 pruned variants and 913 haploids were subjected to PCA. The NA18138 CHD haploids were excluded from the African-Asian (Supplementary Fig. 11b) and European-Asian analyses (Supplementary Fig. 11c) due to apparent inconsistency. For jg1a, jg1b, and jg1c assemblies and JPT haplotypes (Supplementary Fig. 11d), 10,258 pruned variants and 175 haploids were subjected to PCA. For jg1a, jg1b, and jg1c assemblies and Asian populations (Supplementary Fig. 11e), 26,840 pruned variants and 514 haploids were subjected to PCA. For jg1a, jg1b, and jg1c assemblies and worldwide populations (Supplementary Fig. 11f), 23,088 pruned variants and 2037 haploids were subjected to PCA. For PCA of mock JG1 and worldwide populations (Supplementary Fig. 12a–e), 2035 haploids and 23,205, 23,211, 23,206, 23,216, and 23,214 pruned variants were used for the mock JG1 harboring replaced alleles for the SNP sites with $AF \geq 0.5$, 0.6, 0.7, 0.8, and 0.9, respectively.

## SNV validation

*SNV detection by short-read mapping*. The 150-bp paired-end DNBSEQ reads were mapped to the reference hs37d5 using BWA-MEM software. After marking duplicated reads, variants were called using GATK (ver. 4.1.2.0) HaplotypeCaller (normal mode) with the '--pcr-indel-model NONE' option. Examination of whether short-read mapping-based SNV calls supported the genome-by-genome alignment-based SNV calls was performed using the BCFtools software (ver. 1.9) isec command.

*SNV detection by nanopore long-read mapping*. Nanopore super-long and long reads were mapped using minimap2 (ver. 2.17) software with the '--MD -ax map-ont -t 20 -L' options. Variants were ascertained using the Clair software (ver. 2.0.7) callVarBamParallel command with option '--vcf_fn ${KNOWN_VAR-IANTS_VCF} -threshold 0.2', where ${KNOWN_VARIANTS_VCF} indicates the VCF file with genome-by-genome alignment-based SNV variants. We used the pretrained HG002 ONT model "1 + 2 + 2HD + 3 + 4".

## SV analysis (Fig. 3 and Supplementary Fig. 14)

*SV detection*. The JG1, AK1, HX, and ZF1 sequences were aligned to GRCh38 and JG1, AK1, and HX1 sequences were aligned to ZF1 using minimap2 (ver. 2.12) with the '-t 24 -cx asm5 --cs=long' option. The resulting PAF file was subjected to variant calling using the paftools call command. The VCF file was normalized using the BCFtools (ver. 1.8) norm command with the '--threads 4 --multiallelics -both' option. SVs ≥ 51 bp were subjected to the downstream analyses. The number of shared and unique SVs was determined using the SURVIVOR software[61] (ver. 1.0.6) merge command with options '1000 1 1 0 0 1', which indicates regarding two SVs with a distance of both breakpoints ≤ 1000 bp as identical, taking SV type into account. For JG1, SVs detected by comparing the same chromosomes of GRCh38 and JG1 were considered.

*Average depth analysis*. Two hundred Japanese individuals (100 males and 100 females) were selected from the 3552 samples[41]. The 162-bp paired-end reads were mapped using BWA-MEM software as described[41]. Next, accessible regions were defined as the regions where the average depth among the 200 individuals was ≥ 5 and ≤ mean + 2SD; mean and SD were computed for each chromosome. SVs detected within the accessible regions and detected by comparing the same autosomes of GRCh38 and JG1 were considered. The mean value of the average depth within the adjacent upstream region of the SV was regarded as the reference value, and the Δ average depth was defined as the difference between the reference value and the value of the position showing the maximum absolute difference within the SV region.

*Mapping-based SV detection*. PacBio long reads were mapped using NGMLR software (ver. 0.2.7) to detect SVs using Sniffles software (ver. 1.0.11) with default settings. Nanopore super-long reads were mapped using minimap2 software (ver. 2.17) to detect SVs using Sniffles software (ver. 1.0.11) with default settings. The proportion of supported SVs was calculated using the SURVIVOR software[61] merge command with option '1000 1 1 0 0 1'.

## Identification of non-reference sequence

Unmapped reads against the reference GRCh38 from 1070 BAM files were extracted using SAMtools software (ver. 1.9) view command with option '-f 12 -F256' and the bam2fq command. The extracted unmapped reads were re-mapped to JG1 using BWA-MEM software (ver. 0.7.17) with option '-t 12 -K 1000000'. After filtering the BAM output file with MAPQ ≥ 20, the non-reference sequences in JG1 were defined as those with non-zero depth in a 500-bp window plus 500-bp upstream and downstream flanking sequences. These sequences were queried using BLAST (ver. 2.10.0 + ) software against the nt database (downloaded from ftp.ncbi.nlm.nih.gov/blast/db on 12 Feb. 2020).

## Annotations

*Lift over*. GENCODE (ver. 29) annotations on GRCh38 were lifted over to the JG1 coordinates using an in-house script (liftover-gencode2.py). The chain files, which were required for lifting over, were generated from the results of minimap2 between JG1 and GRCh38 using an in-house script (minimap2-to-liftover-chain. py). The two scripts are provided as Supplementary Software.

*De novo gene prediction*. AUGUSTUS[62] (ver. 3.3) were used for gene prediction with '--strand=both --genemodel=complete --singlestrand=true --gff3=on --softmasking=on --species=human' option on the reference genomes with repetitive sequences soft-masked by RepeatMasker software (ver. 4.0.7) with '-species human -xsmall' option after capitalizing all nucleotide letters. Comparison of predicted results and GENCODE dataset was performed using gffcompare software (ver. 0.11.6).

## Rare disease exome analysis (Fig. 4)

Exome analyses were carried out following the GATK best practices for germline variant detection. Short reads were mapped using BWA-MEM software, and the resulting alignment files were sorted and duplication-marked using SAMtools[51] software. Variants of the disease cohort families were called using GATK software (ver. 4.0 to 4.1), and the joint calling process was carried out with samples from other Japanese subjects with various rare diseases. The BED files describing the exome capture regions (SureSelect Human All Exon V4, V5, and V6 Agilent) were lifted over using the paftools liftover command or CrossMap software[63]. The GATK resource bundles were lifted over to the JG1 coordinates using the Picard tools LiftoverVcf command. The SnpEff database[57] was constructed using the lifted-over GENCODE annotation file and used for variant effect predictions. Variants called against JG1 were lifted over to the hs37d5 coordinates using the Picard tools LiftoverVcf command. Overlap relationships between the variants were assessed using the BCFtools (ver. 1.9) isec command.

*Sanger sequencing and validation*. Candidate disease-causing variants in genomic DNA were analyzed by Sanger sequencing[46]. Sequences and Phred quality scores were analyzed using TraceTuner software (sourceforge.net/projects/tracetuner/).

## Allele frequency comparison

The 162-bp paired-end reads from 1070 Japanese individuals were mapped to JG1 or to hs37d5 using BWA-MEM software, and variants were called following GATK best practices[41] with the modification of applying base-quality score recalibration using GATK software (ver. 3.7.0). Variants in bi-allelic SNV sites with the following conditions were selected: (1) 'FILTER = PASS', (2) variants within an accessible region, defined as the range between 0.5× mean depth and 2.0× mean depth, and (3) variants > 1 Mb apart from gap regions. Variant sites matching between JG1 and hs37d5 were identified using transanno software (github.com/informationsea/transanno).

## Statistical tests and graph drawing

Statistical tests were performed using R software (ver. 3.5.1). Histograms were drawn using R software (ver. 3.5.1) and ggplot2 library (ver. 3.0.0).

## Reporting summary

Further information on research design is available in the Nature Research Reporting Summary linked to this article.

## Data availability

The JG1's 624 sequences were deposited to the DNA Data Bank of Japan (DDBJ) under accession number AP023461 (https://getentry.ddbj.nig.ac.jp/getentry/na/AP023461)–AP024084. The raw sequence reads and optical maps were deposited to the DDBJ's Japanese Genotype-phenotype Archive (JGA) under accession number JGAD000362. DDBJ's BioProject and BioSample accession numbers are PRJDB10452 and SAMD00243993, respectively. The JG1 sequence, chain files and GENCODE annotation files are available from the jMorp website. Other datasets are available from Zenodo repository[64]. Raw data for the following figures are available as a Source Data: Figs. 1a, 2a, 2b, 3d, 4a, 4b, Supplementary Figs. 10, 11, 12. Databases used were as follows: SnpEff GRCh38.86 database can be downloaded by the SnpEff download command as "java -jar SnpEff.jar download GRCh38.86"; Blast nt database: ftp://ftp.ncbi.nlm.nih.gov/blast/db UniSTS database ftp://ftp.ncbi.nih.gov/pub/ProbeDB/legacy_unists/; GENCODE database. Source data are provided with this paper.

## Code availability

Scripts are available from the Github repository (https://github.com/juntkym/JG1-paper)[65].

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

## Acknowledgements

This work was supported in part by the Tohoku Medical Megabank (TMM) Project from the Ministry of Education, Culture, Sports, Science and Technology (MEXT) and the Reconstruction Agency; the Japan Agency for Medical Research and Development (AMED; Grant Numbers JP20km0105001 and JP20km0105002) for Tohoku University. This work was also supported in part by JSPS KAKENHI Grant Numbers JP19H05200 and JP19K06625. All computational resources were provided by the ToMMo super-computer system (http://sc.megabank.tohoku.ac.jp/en), which is supported by Facilitation of R&D Platform for AMED Genome Medicine Support conducted by AMED (Grant Number JP20km0405001). We appreciate all the volunteers who participated in the TMM project.

## Author contributions

J.T., S.T, K.Y., C.G., T.F., S.M., S.S., and Y.O. performed computational analyses. J.T., A.K., S.K., and G.T. interpreted the results of rare disease re-analyses. F.K., J.K., A.O., and J.Y. designed and conducted experiments. J.T. and G.T. wrote the manuscript with the assistance of the others. M.S.-Y., K.K., M.Y., and G.T. conceived and supervised the project. All authors read and approved the final manuscript.

## Competing interests

The authors declare no competing interests.

**Additional information**

