## [Peer Review File · Nature Communications]

Reviewers' comments:

Reviewer #1 (Remarks to the Author):

The manuscript presented a new chromosome-level reference genome for Japanese by integrating three *de novo* assemblies. The authors implemented the *de novo* assembly by combining long-read sequencing data, Bionano optical map and short-read Illumina data. To eliminate the bias of rare reference alleles, they *de novo* assembled three genomes of three Japanese individuals and integrated them into a haploid assembly using a meta-assembly strategy. Additionally, authors assembled the genome to a chromosome level by anchoring scaffolds using the genetic and RH maps. Finally, the authors examined practicability of the reference genome by analyzing the exonic variants with rare diseases.

In my opinion, this is a valuable work for building a population-specific reference genome. Considering the complexity of genetic variation, there is no doubt that one reference genome is inadequate to represent human genetic diversity. Besides, the model of discovering variants would be more value by the *de novo* assembly of the target population than aligning to a reference genome. The new assembly JG1 in this study, according to the assembly stat, seems to have substantial improvements of genome quality compared to the exist reference genome and more representativeness for Japanese population. However, for a new assembly genome, though the authors did make some efforts, there are still more quality assessments need to be done before it is endowed as a high-quality reference genome. At the same time, there are several critical technical issues and improper descriptions that need to be addressed.

Major points:

1. In the manuscript, the authors used multiple versions of the human reference genome, such as GRCh38, hg19, GRCh37 or hs37d5. What was the reason of using so many versions? There are a lot of differences between these versions, and even for the same version in different databases, e.g. hg19, GRCh37, and hs37d5. Given the GRCh38 (the latest version is GRCh38.p13) is the highest quality and widely-used version, the authors should unify the versions into the GRCh38.

2. The assessment of assembly quality is insufficient. Routinely, assembly completeness and accuracy need to be assessed. For genome completeness, total bases, the gap (N-bases) number, distribution of gap and gap length are key quality statistics for assembly, but some of them (especially gap stat) were ignored in the manuscript. In particular, the author should conduct a gap filling for reference genome GRCh38 using the JG1, which is also an important index for evaluating the assembly quality. Similarly, for the consensus quality, this manuscript also missed some important evaluations. For example, when the authors compared the JG1 with GRCh38, a numeric percentage is needed to describe the consensus quality between each other, other than a text description that how a high similarity was observed (Line 256). For sequence quality, the author need to calculate the base-error rate of the entire assembly base as presented in a recent genome paper (Ouzhuluobu et al. 2019 National Science Review). It is a an important index to indicate the base accuracy of the assembled genome.

3. The authors compared their genome assembly to a relatively old data (HX1 and AK1). There are more genome assemblies released recently, e.g. thirteen assemblies with global ancestries (Audano et al. 2019 Cell) and ZF1 (a Tibetan individual; Ouzhuluobu et al. 2019 National Science Review). The authors should compare their assembly against these newly assembled genomes. Especially, when implementing the population PCA to detect an "Asianness" pattern, these newer assemblies need to be included (Line 270).

4. I doubt that the way in analyzing major allele substitution for multi-allelic sites, which were chosen randomly without any accordance. For example, if you choose the lowest quality allele (or even the assembled error base) as the final major allele of the reference genome by chance, it will bring a fatal

mistake for variant calling of downstream analysis. In my opinion, a scoring by base quality ranking should be a better option at least.

5. For error correction with short reads, although the authors implemented error-correction using Pilon software, the problematic variants still remain (e.g. indels: premature truncating variants, PTVs) according to Kronenberg et al. (2018 Science). To improve the base quality of the assembly, the authors need to consider these false PTV errors and corrected them. A ready-made pipeline can be easily downloaded from the published study.

6. To identify the structural variants (SVs) between JG1 and GRCh38, the authors adopted an assembly comparison strategy using their assembly sequences. However, considering that the assembly tools are usually not ideal which lead to lots of mistakes. The potential mis-assembling contigs (or scaffolds) might directly affect the results of SV calling. A read-mapping approach is better for SV calling when the deep and long-read sequences are available, which aligns long reads to the reference genome and identifies SVs based on this mapping result (Sedlazeck et al. Nature Methods-2018). Besides, the approach can report SV types, including insertion, deletion, duplication and inversion.

7. In SV analysis, the authors mapped short reads of 200 Japanese to JG1 and GRCh38 and called the SVs, respectively. Then, they compared the difference of read depth between SV regions and the flanking regions and found the difference are smaller in JG1 than in GRCh38, and they concluded that JG1 covers more Japanese population shared SVs and is a better reference genome for Japanese population. However, the author overlooked the difference of assembly strategy can also lead to the difference of SV detection. The JG1 is assembled by long reads but GRCh38 is not, which may bring more SV calling difference caused by mis-assembly regions in GRCh38. To make the conclusion more persuasive, I suggest more long-read assemblies to be included in SV assessment, including the published Asian long-read assemblies (AK1, HX1 and ZF1 etc).

8. The authors only examined exomes when evaluating the practicability of the assembly. However, the variants in noncoding regions have been shown to be associated with common and rare diseases.

9. I don't quite understand how the authors annotate the assembly in this study. Is it predicted by the assembled sequences? Or just lift over the coordinates according to the GENCODE? Additionally, which version of GENCODE was used for annotation? Given the JG1-based exome analysis used the hs37d5 reference, if the authors want to make the coordinates consistent, the GRCh37 version of GENCODE should be used. But GRCh37 is the old version of human reference annotation and you may miss the key updated annotations. Please clarify.

10. Genomes usually contain non-repetitive sequences that are missing from the reference genome and occur only in a population subset. Such non-reference non-repetitive (NRNR) sequences are a strong support for generating a population-diversity reference genome. It is recommended that the authors need to evaluate the NRNR sequences in JG1.

11. The authors did not provide any experimental validation for the called variant sets,

Minor points:

1. For the donor individuals, more detail information need be provided, especially the healthy status and age etc.

2. Supplement Figure 2, according to the dot plots, the inversion (p12q13) occurs in jg1b and jgc1 other than jg1a. However, in the main text (Line 161), the authors said it is jg1a who has this inversion. Please clarify.

3. There are too many numeric information presented in Table1. For example, the stat of jg1a,b,c can be moved to supplementary materials.
4. Line 263, why the used reference genome changed to hg19? Please provide the reason.
5. Line 278, again, please provide reason why the used reference genome was changed to hs37d5?
6. Line 303, the author just presented the SVs between 50bp-10kbp in length. Was this due to limitation by tools or data? How about the situation of longer SVs?
7. Line 347, the authors claimed that JG1 has fewer false-positives than the reference genome when detecting variants. I think it is ambiguous using "false-positives" here. Does it mean the false variant? (any validation support?), or it just refers to the non-population specific variants? Or the disease casual variants? Please provide further explanation.
8. Line 364, I think it is premature to use "several advantages" here. The authors didn't present "several advantages" of JG1 as a new reference. Please rephrase here or put more examples of the claimed advantages.
9. Line 395, a phased genome is better for characterizing the variants, which can increase nearly 30% SV calls than a un-phased genome (especially for heterozygous variants). It would be better if the authors can conduct phasing.

Reviewer #2 (Remarks to the Author):

The authors first describe the construction of JG1 reference genome from de novo assembly of PacBio long reads combined with Bionano Genomics' optical maps, followed by pseudo-molecule construction based on genetic and radiation hybrid maps, and then demonstrate the utility of JG1 as a Japanese reference genome. This work is well organized and very comprehensive.

Here're minor comments:

1. The 2nd sentence from line 47 went wrong.
2. The exact name of the company is Bionano Genomics. Therefore, its full name should be given first at line 110 as done for Pacific Bioscience at line 109 for consistency.
3. As long as I know, Bionano raw data are called optical maps but their de novo assembled maps are called genome maps, which are used for hybrid scaffolding with PacBio contigs.
4. The meaning of the sentence starting with "Although assemblies" at line 117 is a bit confusing. In fact, many good de novo assemblies are not organized into pseudo-molecules limiting their utility as a reference genome. In this sense, I think it's better to use "pseudo-molecules" instead of "a set of chromosomes".
5. Around line 269, at first glance, it appears to claim that JG1 is NOT localized within the Asian cluster but associated with the Japanese cluster. This might have been caused by uncertainty of PCA interpretation due to small variances assigned to the major two components. So the variances explained should be given in the PCA plot.
6. In line 315, 'piling-up' is not an indication of an insertion but of a copy gain. It's difficult to detect an insertion with unique sequence, purely based on read depth.

7. The description of Mate-pair sequencing from line 450 doesn't make sense. The sizes of mate pair libraries are 500bp on average?

8. It is unclear why "amplification" should be used when it's just electronic PCR at line 576. Please clarify the meaning.

Responses to Reviewer #1:

We sincerely appreciate the reviewer's thorough reading, helpful comments, and evaluation. We have addressed all of the concerns raised by the reviewer. The reviewer's comments are written in blue, whereas our responses are in black, and revised text in the manuscript is red.

Reviewer #1 (Remarks to the Author):

The manuscript presented a new chromosome-level reference genome for Japanese by integrating three *de novo* assemblies. The authors implemented the *de novo* assembly by combining long-read sequencing data, Bionano optical map and short-read Illumina data. To eliminate the bias of rare reference alleles, they *de novo* assembled three genomes of three Japanese individuals and integrated them into a haploid assembly using a meta-assembly strategy. Additionally, authors assembled the genome to a chromosome level by anchoring scaffolds using the genetic and RH maps. Finally, the authors examined practicability of the reference genome by analyzing the exonic variants with rare diseases.

In my opinion, this is a valuable work for building a population-specific reference genome. Considering the complexity of genetic variation, there is no doubt that one reference genome is inadequate to represent human genetic diversity. Besides, the model of discovering variants would be more value by the *de novo* assembly of the target population than aligning to a reference genome. The new assembly JG1 in this study, according to the assembly stat, seems to have substantial improvements of genome quality compared to the exist reference genome and more representativeness for Japanese population. However, for a new assembly genome, though the authors did make some efforts, there are still more quality assessments need to be done before it is endowed as a high-quality reference genome. At the same time, there are several critical technical issues and improper descriptions that need to be addressed.

Again, we appreciate the reviewer's evaluation of the value of this work for building a population-specific reference genome. We have addressed all of the concerns raised by the reviewer as follows.

Major points:

1. In the manuscript, the authors used multiple versions of the human reference genome,

such as GRCh38, hg19, GRCh37 or hs37d5. What was the reason of using so many versions? There are a lot of differences between these versions, and even for the same version in different databases, e.g. hg19, GRCh37, and hs37d5. Given the GRCh38 (the latest version is GRCh38.p13) is the highest quality and widely-used version, the authors should unify the versions into the GRCh38.

The reviewer has pointed out the lack of a reason why we used different versions of the reference genome for different analyses and recommended that we use the latest major release version, GRCh38. The versions we used were hg19 for the PCA **(a)**, hs37d5 for SNV analysis **(b)**, hs37d5 also for exome analyses **(c)**, and GRCh38 for SV analyses **(d)**.

For PCA **(a)**, we changed the reference genome from hg19 to GRCh38 in the revised version (in new **Figure 2a**) according to the reviewer's suggestion. We also revised the sentence referring the PCA, which now reads as follows:

Page 11, line 293: To assess whether JG1 is a representative reflection of the SNV composition of the Japanese population, we performed PCA using JG1, the reference **GRCh38, 13 assemblies from diverse populations, and 2,022 haplotypes constructed from 11 HapMap3 populations (see Methods section and Supplementary Tables 6 and 9 for the derived population of the assemblies).**

The purpose for the SNV analysis **(b)** was to compare the allele frequency (AF) of the detected SNVs between JG1 and the reference. The most comprehensive AF panel in the Japanese population was built using hs37d5 (Tadaka et al., 2019), and hence, we chose hs37d5. We added a sentence to explain this rationale and changed the notation "non-hg19-type allele" to "non-hs37d5-type allele" for simplicity, which now reads as follows:

Page 12, line 311: To assess whether JG1 harbors the major allele among the Japanese population across SNV sites, we aligned JG1 against the reference genome hs37d5, detected SNVs, and investigated their allele frequency (AF) using the AF panel of **3,552 Japanese individuals (namely, the 3.5KJPNv2 AF panel⁴¹).** We chose hs37d5 as the reference because the 3.5KJPNv2 AF panel was built on that reference genome; **hs37d5 is a reference genome based on GRCh37/hg19 primary sequences amended mainly with those of other high-quality assemblies, and it is thus well-suited and widely used for NGS analyses⁴².**

Page 13, line 325: ... , in which the horizontal axis indicates the non-hs37d5-type allele ...

For the exome analysis (c), we used the SureSelect exome capture kit, in which the capture region was designed based on the reference GRCh37. Because hs37d5 is an optimized version of GRCh37 for NGS analysis (The 1000 Genomes Project Consortium, 2012), we chose hs37d5 for the exome analysis. We added a new sentence to explain this rationale, which now reads as follows:

Page 16, line 425: For comparison, we also performed exome analyses using hs37d5 as the reference; we used hs37d5 because the exome capture sequence was designed using the GRCh37/hg19 sequence.

For the SV analysis (d), there were no large-scale reference SV panels for the Japanese population, and hence, we used the most recent major release version of the reference, GRCh38. We note that the difference between GRCh38 and the later minor release versions such as GRCh38.p13 is in the patch sequences that are either FIX contigs or ALT contigs, both of which cannot be used with the primary sequences in the standard mapping approach; the standard mapping approach assumes the reference genome as haploid, and the primary sequences are the same between GRCh38 and the later minor release versions. Therefore, the minor release versions are not relevant for this context. We added a sub-section in the Methods section that explains this rationale, which reads as follows:

Page 30, line 779: Assembly assessment

Choice of the reference genome: GRCh37/hg19 and GRCh38/hg38 are the two most widely used reference genomes. For analyses using short-read mapping, we used hs37d5, an optimized GRCh37 for NGS analysis. For other analyses, we used an analysis-ready version without ALT contigs of the latest reference genome, GRCh38, which was downloaded from the Illumina iGenome website (ftp://ussd-ftp.illumina.com/Homo_sapiens/NCBI/GRCh38/Homo_sapiens_NCBI_GRCh38.tar.gz).

2. The assessment of assembly quality is insufficient. Routinely, assembly completeness and accuracy need to be assessed. For genome completeness, total bases, the gap (N-bases) number, distribution of gap and gap length are key quality statistics for assembly, but some of them (especially gap stat) were ignored in the manuscript. In particular, the author should conduct a gap filling for reference genome GRCh38 using the JG1, which is also an important index for evaluating the assembly quality. Similarly, for the consensus quality, this manuscript also missed some important evaluations. For example, when the authors compared the JG1 with GRCh38, a numeric percentage is needed to

describe the consensus quality between each other, other than a text description that how a high similarity was observed (Line 256). For sequence quality, the author need to calculate the base-error rate of the entire assembly base as presented in a recent genome paper (Ouzhuluobu et al. 2019 National Science Review). It is a an important index to indicate the base accuracy of the assembled genome.

The reviewer advised us to perform more comprehensive quality assessments of JG1, which can be summarized as four specific points: **(a)** present assembly evaluation statistics including total bases, the gap number, distribution of gaps and gap length, **(b)** perform gap-filling of the reference GRCh38 using JG1, **(c)** provide numeric percentages of consensus quality between GRCh38 and JG1, and **(d)** provide the base-error rate of the entire assembly. According to the reviewer's comment, we revised our manuscript as follows.

For assembly quality statistics **(a)**, we provided the total bases, gap number, and total gap length in revised **Table 1** and new **Supplementary Table 4**. For the distribution of gaps, we described an idiogram for JG1 (**Figure 1b**) and added Circos plots for individual draft assemblies and for JG1 in new **Supplementary Fig. 6**. We added following text in the Results section:

Page 8, line 194: The number and length of gap regions in the hybrid scaffolds were 417, 413, and 380 and 34.3 Mb, 28.3 Mb, and 24.5 Mb, respectively (Supplementary Table 4; see Supplementary Fig. 6a–c for the gap position).

Page 10, line 269: The total length of JG1 was approximately 3.1 Gb, including 473 gap regions of 251 Mb in total length, of which 227 Mb was intentionally inserted to represent telomeric, centromeric, and heterochromatic regions (Table 1, Figure 1b, Supplementary Fig. 6d, and Supplementary Table 8).

For the gap-filling ability of JG1 **(b)**, we performed gap-filling of GRCh38 using the uniline software described by Shi et al. (2016). We found that JG1 successfully filled the largest number of gaps uniquely among other assemblies (summarized in new **Supplementary Table 9**). We added the following text:

Page 11, line 287: Moreover, we assessed whether JG1 can fill the remaining gaps in the reference GRCh38²⁴, and found that JG1 uniquely filled 36 gaps, which was the second highest number of uniquely filled gaps among other assemblies (Supplementary Table 9).

For the evaluation of consensus quality (**c**), we quantified the average identity and covered reference region using dnadiff software, and the numerical percentage for consensus quality was added to the text, which reads as follows (new **Supplementary Table 9** provides a comparison with other assemblies):

Page 11, line 277: We also quantified consensus quality using dnadiff software⁴⁰ and found that JG1 covered 95.53% of the reference with 99.79% average identity (Supplementary Table 9).

For the base-error rate (**d**), we assessed the base-error rate for individual draft assemblies (polished hybrid scaffolds) using the method described by Ouzhuluobu et al. (2019). The base-error rate was 0.0010%, 0.0012%, and 0.0015% for the jg1a, jg1b, and jg1c assemblies, respectively, considerably lower than the 0.01% criterion. Note that this method of base-error estimation is not applicable to JG1 because JG1 is a meta-assembly of three individual assemblies with its minor allele substituted with the major allele. However, because JG1 is made up of three assemblies, the base-error rate would be in the range of the three. We provided these results in new **Supplementary Table 7** and added the following sentence:

Page 8, line 198: The estimated base-error rate of the three sets of polished hybrid scaffolds was $1.02\text{--}1.46 \times 10^{-5}$, being well below the standard base-error rate of 1×10^{-4} for reference quality²⁵ (Supplementary Table 7).

We also provided methods for the above assessments in the Methods section on page 30–31.

3. The authors compared their genome assembly to a relatively old data (HX1 and AK1). There are more genome assemblies released recently, e.g. thirteen assemblies with global ancestries (Audano et al. 2019 Cell) and ZF1 (a Tibetan individual; Ouzhuluobu et al. 2019 National Science Review). The authors should compare their assembly against these newly assembled genomes. Especially, when implementing the population PCA to detect an “Asianness” pattern, these newer assemblies need to be included (Line 270).

As suggested by the reviewer, the results of comparisons of assembly-evaluation statistics versus HX1, AK1, ZF1, and the thirteen assemblies are provided in revised **Table 1** and new **Supplementary Tables 6** and **9**. Revised **Table 1** and new **Supplementary Table 6** provide basic assembly statistics, and new **Supplementary Table 9** describes other

evaluations, including consensus quality, protein-truncating variants (PTVs), and gap-filling ability for all of the above assemblies.

PCA with additional assemblies is provided in revised **Figure 2a** and **2b**. In the PCA, CHM1 and CHM13 were omitted because of their unknown ancestries, and NA12878, NA19240, and NA19434 were omitted because the HapMap3 samples used for PCA already included haplotypes of their parents or siblings; inclusion of such close relatives is not assumed for PCA.

We also described the method in the Methods section on page 32–33.

4. I doubt that the way in analyzing major allele substitution for multi-allelic sites, which were chosen randomly without any accordance. For example, if you choose the lowest quality allele (or even the assembled error base) as the final major allele of the reference genome by chance, it will bring a fatal mistake for variant calling of downstream analysis. In my opinion, a scoring by base quality ranking should be a better option at least.

The reviewer raised concerns regarding our majority decision strategy, especially in terms of randomly choosing an allele for the multi-allelic sites, pointing out the possibility of choosing an erroneous base at those sites, and suggested a method to choose an allele according to the base-quality ranking among the three assemblies.

To address these concerns, we first assessed how many erroneous bases were incorporated during the random choice at the multi-allelic sites. The total number of multi-allelic sites was 120,937 (1,139 SNV–SNV sites, 11,220 SNV–indel sites, and 108,578 indel–indel sites), and among these, only 30 alleles, which were adopted as the new reference allele after random choice (6 SNV alleles and 24 indel alleles), were estimated as being erroneous by base-error rate estimation (in new **Supplementary Table 7**). Although we admit that the random choice strategy cannot prevent all erroneous bases from being chosen, the number of such sites was just 30 per 2.86 Gb (0.000001%), or 0.025% of all multi-allelic sites, which we think is acceptable. We added the discussion above as a **Supplementary Note** and in the Results section, which now reads as follows:

Page 9, line 227: The total number of multi-allelic sites was 120,937, among which it was estimated erroneous alleles were chosen at only 30 sites via this random choice (see Supplementary Note for more details).

Moreover, as described in the next response, we addressed how many PTVs exist in JG1 and found that the number of PTVs in JG1 was the lowest among the other high-quality reference genomes; even lower than that of the hydatidiform mole assemblies (in new **Supplementary Table 9**). This comparison demonstrates the efficacy of our majority decision strategy to remove as many erroneous bases as possible.

As for base-quality ranking, there are no established methods, as far as we know, to evaluate base-quality in a base pair-wise manner for an assembled genome, such as those producing final assembly in, for example, the FASTQ format, which would be an important research field. However, even if such a method existed, the base-quality ranking method would produce bias toward choosing alleles from the most accurately assembled genome among the three individuals, which is not relevant to whether the allele is common or rare in the Japanese population (i.e., rare or even private alleles in the most accurate assembly are more likely to be chosen). Because our purpose was to remove as many private variants derived from a single individual as possible, this strategy would result in a genome harboring more private variants from a single subject, and hence, would not fit our purpose.

5. For error correction with short reads, although the authors implemented error-correction using Pilon software, the problematic variants still remain (e.g. indels: premature truncating variants, PTVs) according to Kronenberg et al. (2018 Science). To improve the base quality of the assembly, the authors need to consider these false PTV errors and corrected them. A ready-made pipeline can be easily downloaded from the published study.

The reviewer recommended quantifying the number of problematic variants in JG1 and correcting them using the method described by Kronenberg et al. (2018). We first counted the problematic variants that cause premature protein truncation due to missense, frame-shift, or splice-site disruption by annotating the impact of all of the detected variants and found that JG1 harbors the lowest number of PTVs among other reference-quality assemblies (given in new **Supplementary Table 9**). We also provided the method used to count PTVs in the Methods section on page 31.

Moreover, we carefully examined the indel correction pipeline described by Kronenberg et al. (2018) and in their GitHub repository (github.com/EichlerLab/indel_correction_pipeline) and found that the pipeline performs (1) indel-specific error correction based on the short-read mapping-based variant callings, (2) genome-by-genome alignment of the draft assembly to the reference GRCh38/hg38 to identify protein-truncating indels using gene annotation on the reference, and (3) selective correction of problematic indels on the draft assembly. Therefore, the pipeline uses the information annotated to the reference genome GRCh38/hg38, and thus, the resulting corrected assembly cannot be regarded as independent from the reference. Note that lifting over gene annotations from the reference itself is not problematic if the purpose is to annotate the genome, but it is problematic if the purpose is to determine the sequence of the assembly and call it independent.

Nevertheless, we applied the indel correction pipeline to the three individual assemblies and found that it removed approximately 200 PTVs (in new **Supplementary Table 18**), which is comparable to our majority decision strategy. Given that using the pipeline violates the notion of independent assembly and the original version of JG1 has the lowest number of PTVs, we believe that the presented version of JG1 already deserves reference-quality assembly. We added the discussion above in the Discussion section, with reference to new **Supplementary Table 18**, and this text reads as follows:

Page 18, line 480: The accuracy of JG1 was comparable or superior to other high-quality genomes especially in terms of the number of PTVs (Supplementary Table 9). The remaining PTVs, however, might be removed by a genic region-specific indel error-correction pipeline⁴⁸. Although application of the pipeline to our data reduced PTVs in the individual assemblies (Supplementary Table 18), careful examination of the pipeline revealed that it requires information from the reference GRCh38/hg38. Since our purpose was to construct a genome assembly completely independent of the reference genome, we did not adopt this pipeline. Nonetheless, because our majority decision strategy among the three assemblies was as effective as the pipeline, the current version of JG1 deserves the reference quality assembly.

6. To identify the structural variants (SVs) between JG1 and GRCh38, the authors adopted an assembly comparison strategy using their assembly sequences. However, considering that the assembly tools are usually not ideal which lead to lots of mistakes. The potential mis-assembling contigs (or scaffolds) might directly affect the results of SV

calling. A read-mapping approach is better for SV calling when the deep and long-read sequences are available, which aligns long reads to the reference genome and identifies SVs based on this mapping result (Sedlazeck et al. Nature Methods-2018). Besides, the approach can report SV types, including insertion, deletion, duplication and inversion.

The reviewer recommended that we detect SVs by mapping long reads to gauge potential false-positive calls using genome-by-genome alignment due to possible mis-assemblies. To address this, we mapped PacBio long reads for the three subjects to the reference GRCh38, called SVs by the suggested method, Sniffles (Sedlazeck et al., 2018), and compared them to the alignment-based SV calls. We found that 74.8% of insertion calls and 81.3% of deletion calls were supported by mapping-based SV calls from at least one individual. We summarized the SV analysis results including the other types of SVs in new **Supplementary Table 13** and **Supplementary Fig. 11**.

7. In SV analysis, the authors mapped short reads of 200 Japanese to JG1 and GRCh38 and called the SVs, respectively. Then, they compared the difference of read depth between SV regions and the flanking regions and found the difference are smaller in JG1 than in GRCh38, and they concluded that JG1 covers more Japanese population shared SVs and is a better reference genome for Japanese population. However, the author overlooked the difference of assembly strategy can also lead to the difference of SV detection. The JG1 is assembled by long reads but GRCh38 is not, which may bring more SV calling difference caused by mis-assembly regions in GRCh38. To make the conclusion more persuasive, I suggest more long-read assemblies to be included in SV assessment, including the published Asian long-read assemblies (AK1, HX1 and ZF1 etc.).

According to the reviewer's recommendation, we called SVs in the published Asian assemblies AK1, HX1, and ZF1 using genome-by-genome alignment and compared the results with those for JG1 (given as new **Supplementary Fig. 12a**). The number of individual assembly-specific SVs and shared SVs was similar among the four assemblies, except for HX1, which had a lower number of SVs. This might have resulted from the fact that the HX1 assembly is a bit more fragmented than the other three assemblies, which results in a shorter callable region and fewer variant calls. We provided these analyses and interpretations in the Results section (see below).

We also performed genome-by-genome alignment-based SV calling employing the Tibetan reference ZF1 as a reference genome (given as new **Supplementary Fig. 12b**). We found that the number of SVs in the Asian assemblies called against ZF1 was lower than that against GRCh38. This reduction might have resulted from the fact that the ancestry of the Asian assemblies is closer to ZF1 than to GRCh38 or because the Asian assemblies were built using the same assembly strategy. We described these analyses as below and in new **Supplementary Fig. 12**:

Page 14, line 367: We also compared the detected SVs in JG1 with the other three Asian assemblies: AK1, HX1, and ZF1. Genome-by-genome alignment against GRCh38 detected comparable numbers of both insertions and deletions among the four assemblies, with the exception of slightly lower number in HX1, due probably to its fragmented assembly (Supplementary Fig. 12a). We also performed genome-by-genome alignment-based SV detection using ZF1 as a reference (Supplementary Fig. 12b) and found that fewer insertions and deletions were detected in JG1 compared with using GRCh38 as the reference (4,718 vs 8,697 insertions and 5,341 vs 6,190 deletions against ZF1 vs GRCh38, respectively), probably due to their ancestral closeness and/or similarity in assembly strategies.

8. The authors only examined exomes when evaluating the practicability of the assembly. However, the variants in noncoding regions have been shown to be associated with common and rare diseases.

Common and rare disease-causing variants in non-coding regions can be assessed by whole-genome sequencing (WGS) analysis using JG1. To evaluate the applicability of JG1 to WGS analysis, we mapped WGS reads of 1,070 Japanese individuals to JG1 and compared the allele frequency called from the same short-read dataset mapped to the reference hs37d5. We found that the frequency of non-hs37d5-type alleles between the two references was consistent over a broad range of frequency ($R > 0.999$). The allele frequency comparison is provided in new **Supplementary Fig. 14**, and we added following text:

Page 17, line 458: We also assessed whether JG1 can serve as the reference for whole genome sequencing (WGS) analysis by mapping WGS short reads of 1,070 Japanese individuals to JG1 and comparing the non-GRC-type AF with those of the reference hs37d5. We found that the non-GRC-type AF was almost equivalent between the two

references throughout the wide allele frequency range ($R > 0.999$) for the 19,772,783 SNV sites called in autosomes of both references (Supplementary Fig. 14).

9. I don't quite understand how the authors annotate the assembly in this study. Is it predicted by the assembled sequences? Or just lift over the coordinates according to the GENCODE? Additionally, which version of GENCODE was used for annotation? Given the JG1-based exome analysis used the hs37d5 reference, if the authors want to make the coordinates consistent, the GRCh37 version of GENCODE should be used. But GRCh37 is the old version of human reference annotation and you may miss the key updated annotations. Please clarify.

The reviewer pointed out the lack of a description of the genome annotation methods and raised concerns regarding lifting over annotations from an old database.

With respect to the annotation method, the JG1 genome was annotated by lifting over GENCODE annotations (ver. 29) from GRCh38. We prepared an independent sub-section for annotation in the Methods section and moved the text regarding the annotation method to this sub-section. In addition, we provided source code for the two in-house scripts in the **Supplementary Note**. The method sub-section reads as follows:

Page 35, line 900: Annotations

GENCODE (ver. 29) annotations on GRCh38 were lifted over to the JG1 coordinates using an in-house script (`liftover-gencode2.py`). The chain files, which were required for lifting over, were generated from the results of minimap2 between JG1 and GRCh38 using an in-house script (`minimap2-to-liftover-chain.py`). The two scripts are provided in the **Supplementary Note**.

For the exome analysis, we lifted over the capture regions of the SureSelect exome capture kit from GRCh37 to JG1, because the exome capture kit was designed according to the coordinates and sequence of the reference GRCh37. We agree that the updated gene annotation is more comprehensive, but the exome capture region must be lifted over from those with matched reference versions. The possible lack of annotation would be less relevant for this analysis because the dataset is a re-analysis of disease-causing variants previously identified using GRCh37/hg19 as the reference.

10. Genomes usually contain non-repetitive sequences that are missing from the reference genome and occur only in a population subset. Such non-reference non-repetitive (NRNR) sequences are a strong support for generating a population-diversity reference genome. It is recommended that the authors need to evaluate the NRNR sequences in JG1.

We thank the reviewer for this insightful suggestion to analyze NRNR sequences in JG1. The original NRNR study (Kehr et al., 2017) collected and assembled short reads unmapped against the reference genome from 15,000 Icelanders to identify NRNR sequences for them. To perform NRNR analysis of JG1, we collected reads unmapped against the reference genome hs37d5 from 1,070 Japanese individuals and re-mapped them to JG1 to identify possible Japanese NRNR sequences. We found 28 such regions (59 kb in sum) using highly stringent criteria. We described these finding in the Results section as follows:

Page 15, line 392: The reference genome might lack some population-specific sequences^{13,14}, thus might make some short reads from samples in the population unmapped. To determine whether JG1 has Japanese population-specific sequences not present in the reference genome, we collected the unmapped reads of 1,070 Japanese individuals⁴⁴ when mapped to the reference hs37d5 and re-mapped them to JG1. Of the $574 \pm 20 \times 10^6$ reads per individual, $4.3 \pm 1.5 \times 10^6$ reads were flagged as unmapped to hs37d5 ($n = 1,070$; mean \pm SD). Of these, we found that $4,820 \pm 1,673$ reads could be successfully mapped to JG1 with high mapping quality ($\text{MAPQ} \geq 20$). These reads were mapped to 2,078 distinct regions in JG1, and 28 regions had at least one mapped read from every individual (Supplementary Table 16). Among the 28 regions, 23 exhibited similarity to previously reported non-reference sequences^{13,14}, 2 exhibited similarity to those included in the next version of the reference (GRCh38), and 1 exhibited similarity to a previously reported sequence from a BAC clone of Yoruban individual NA18507⁴⁵. One of the remaining 2 regions was 2 kb in length and exhibited only 89.7% identity for approximately half of the region (998 bp) with a BAC clone sequence of RP-11, AC124768.4, which is present in the reference hs37d5 in a matched chromosome. The other region, which was 1.5 kb in length and found in chromosome 7, exhibited only 85.5% identity to a chimpanzee BAC clone CH251-285G22 sequence (AC190217.3) from a matched chromosome.

To cite the paper and the other related papers, we modified a sentence in the Introduction section as follows:

Page 4, line 94: In addition, recent **studies** revealed a lack of (population-specific) sequences in the reference genome, and discovered thousands of structural variants (SVs) in world-wide samples¹¹⁻¹⁴.

11. The authors did not provide any experimental validation for the called variant sets,

The reviewer pointed out the lack of experimental validation for the called variant sets. In the manuscript, we described three variant sets: **(a)** SNVs and **(b)** SVs identified by aligning JG1 against the reference genome, and **(c)** the rare disease exome analyses for seven families.

For SNVs **(a)**, we prepared new BGI Genomics DNBseq short reads ($\geq 42\times$ per individual; given as new **Supplementary Table 11**) and Oxford Nanopore Technology (ONT) long reads ($\geq 40\times$ per individual; given as new **Supplementary Table 12**), neither of which were used for constructing JG1. These independent and orthogonal datasets can be used for validation because DNBseq is a short-read sequencing platform independent from the Illumina HiSeq, with different chemistry, and ONT long-read sequencing is based on fundamentally different sequencing technologies. Although the base quality for the ONT long reads was relatively low ($Q \sim 15$; i.e., error rate $\sim 3\%$), the Clair variant caller can achieve 97–99% precision and 93–97% recall with the latest variant-calling model. Mapping and variant calling from the two independent datasets ascertained 93.9–95.9% of the SNVs detected using genome-by-genome alignment. We added the following text:

Page 12, line 320: Of these SNVs, 93.9–95.9% were validated by two independent mapping analyses, namely BGI Genomics DNBseq short read and Oxford Nanopore Technologies (ONT) long-read datasets not used for constructing JG1 (see Supplementary Tables 10–12 and Methods section).

For SVs **(b)**, we performed mapping-based SV detection using the PacBio long reads and ONT super-long reads ($\geq 32\times$ per individual) for orthogonal validation. We found that 74.8-90.2% of insertions and 81.3-86.9% of deletions were supported by mapping-based SV calls from at least one individual (given as new **Supplementary Table 15**). We added the following text in the Results section:

Page 14, line 357: To validate the detected SVs, we performed two orthogonal SV analyses based on mapping the PacBio long reads used for constructing JG1 and ONT long reads not used for constructing JG1 (Supplementary Tables 12–14). We mapped

PacBio and ONT long reads using NGMLR and minimap2, respectively, and detected SVs using Sniffles software⁴³. We found that both mapping-based analyses detected a comparable or larger number of SVs of similar size and chromosomal distribution as well as other types of SVs (Supplementary Tables 13 and 14 and Supplementary Fig. 11), supporting 74.8–90.2% of the detected insertions and 81.3–86.9% of deletions (Supplementary Table 15).

For exome-identified SNVs and indels (c), we conducted Sanger sequencing for the 7 presented families and additional 24 families for whom we also performed JG1-based and hs37d5-based exome analyses as a separate study. We found that both hs37d5 and JG1 exhibited equivalent accuracy: positive predictive value of 98% (57/58 variants) and sensitivity of 97% (57/59 variants) for both JG1 and hs37d5; the 1/58 pseudo-positive variant and the 2/59 pseudo-negative variants were the same variants between JG1 and hs37d5. The results are now summarized in new **Supplementary Table 17**. We added the following text in the Results section:

Page 17, line 439: To validate the variants detected via the exome analysis using JG1 as a reference, we Sanger-sequenced the 8 regions harboring the disease-causing variants of the 22 individuals from 7 families (Supplementary Fig. 13), along with an additional 42 regions of 44 individuals from 24 families. The total Sanger-sequenced length was 25,896 bp, with mean Phred score of 33.9. The sequenced regions corresponded to 50 distinct regions 12,081 bp and 12,088 bp in length for hs37d5 and JG1, respectively. Within these 50 regions, we detected 58 variant calls from both exome analyses, and 57 calls were ascertained by Sanger sequencing for both JG1 and hs37d5, corresponding to a positive predictive value of 98.3% (Supplementary Table 17). The sensitivity was also the same between these two genomes (57/59; 96.6%; Supplementary Table 17). These results suggest that variant calls from JG1 are as accurate as those from the reference hs37d5.

Minor points:

1. For the donor individuals, more detail information need be provided, especially the healthy status and age etc.

We have corrected text in question in the Methods section, which now reads as follows:

Page 20, line 536: Donor selection: Three adult male Japanese volunteers were recruited (45–59 years old) and participated in this study with written, informed

consent. They were self-reportedly healthy without genetic diseases, and had Japanese ancestry.

2. Supplement Figure 2, according to the dot plots, the inversion (p12q13) occurs in jg1b and jg1c other than jg1a. However, in the main text (Line 161), the authors said it is jg1a who has this inversion. Please clarify.

The two inversions are different. The *pericentric* inversion $inv(9)(p12q13)$ in jg1a is not visible in the dot plots because the assembly software and/or sequencing technology has difficulty in that region; hence, there are no successfully assembled contigs or scaffolds covering that region. On the other hand, the 2.6-Mb inversion shared by jg1b and jg1c resides in the *long arm* of chromosome 9. To clarify the above points, we modified the figures by adding a black bar and blue arrows indicating each inversion and also modified the legend for revised **Supplementary Fig. 2**, which reads as follows:

Page 49, line 1175: Supplementary Fig. 2. Harr plot of the alignment between chromosome 9 of GRCh38 and the two largest scaffolds aligned to chromosome 9 from the (a) jg1a, (b) jg1b, and (c) jg1c assemblies, indicating that the $inv(9)(p12q13)$ in jg1a did not appear to affect the assembly and that the two individual genomes (b, c) harbor a possible shared inversion. 'Super-scaffold' is the default prefix designated by BionanoSolve software. The black bar in (a) indicates the position of $inv(9)(p12q13)$. Blue arrows indicate the possible shared 2.6-Mb inversion.

3. There are too many numeric information presented in Table1. For example, the stat of jg1a,b,c can be moved to supplementary materials.

We separated the Table into two: revised **Table 1** now describes the statistics for JG1 and assemblies from other research groups (ZF1, AK1, HX1, Swe1, and Swe2), whereas new **Supplementary Table 4** describes the statistics for intermediate assemblies to construct JG1.

4. Line 263, why the used reference genome changed to hg19? Please provide the reason.

We changed the reference genome to GRCh38 (**Figure 2a**). Note that the later minor release versions are not relevant for this analysis because patch sequences cannot be used with the primary autosomal sequences; only autosomal variants were relevant for this analysis.

5. Line 278, again, please provide reason why the used reference genome was changed to hs37d5?

We used hs37d5 as the reference genome for analyses using short-read mapping. We provided the reason why we used hs37d5 as the reference as follows:

Page 12, line 311: To assess whether JG1 harbors the major allele among the Japanese population across SNV sites, we aligned JG1 against the reference genome hs37d5, detected SNVs, and investigated their allele frequency (AF) using the AF panel of 3,552 Japanese individuals (namely, the 3.5KJPNv2 AF panel⁴¹). We chose hs37d5 as the reference because the 3.5KJPNv2 AF panel was built on that reference genome; hs37d5 is a reference genome based on GRCh37/hg19 primary sequences amended mainly with those of other high-quality assemblies, and it is thus well-suited and widely used for NGS analyses⁴².

6. Line 303, the author just presented the SVs between 50bp-10kbp in length. Was this due to limitation by tools or data? How about the situation of longer SVs?

We newly included the SV calls larger than 10 kb, resulting in the addition of 8 insertions and 13 deletions (given as revised **Figure 3a, 3b, and 3d**). We had omitted those SVs in the original manuscript because such large SVs—especially insertions—are difficult to validate in an orthogonal long-read mapping experiment with a mean read length of ~10 kb and hence less reliable. Although less reliable, we revised the text to include SVs >10 kb while noting the range is less reliable and referring to the validation by mapping analysis (newly provided in **Supplementary Tables 13 and 15** and **Supplementary Fig. 11**), which reads as follows:

Page 13, line 344: A genome-by-genome comparison detected 8,697 insertions and 6,190 deletions >50 bp in length. The largest insertion and deletion were 15,621 bp and 17,221 bp, respectively; 8 insertions \geq 10 kb were less reliable because mapping-based orthogonal validation did not detect ones in that range (see below).

Page 15, line 385: To determine whether this pattern is common among SVs throughout the genome, we compared the maximum difference in average depth between the SV region and its adjacent upstream region of the same length and found that the difference in the average depth was smaller in JG1 than GRCh38 (Figure 3d; $P = 8.8 \times 10^{-11}$ for $n = 3,953$ pairs of insertions; $P < 2.2 \times 10^{-16}$ for $n = 2,767$ pairs of deletions; Wilcoxon signed rank tests).

7. Line 347, the authors claimed that JG1 has fewer false-positives than the reference genome when detecting variants. I think it is ambiguous using “false-positives” here. Does it mean the false variant? (any validation support?), or it just refers to the non-population specific variants? Or the disease casual variants? Please provide further explanation.

We agree that the sentence was confusing. We meant "non–disease-causing variants" by the term false-positives in the original manuscript. However, we re-considered the sentence because whether a variant is disease-causing or not can be addressed only after one examines the patient’s phenotype, which is not relevant to the number of variants per se. Therefore, we thought that "total candidate variants" would be more appropriate in this context. For a rare disease exome analysis, one or two disease-causing variants would usually be identified from the total pool of candidate variants; the others are non–disease causing variants. With respect to experimental validation, almost all of the detected variants were ascertained by Sanger sequencing. We corrected the text in question, which reads as follows:

Page 17, line 436: These findings suggest that JG1 produces fewer **total candidate variants** while successfully detecting disease-causing **variants** in whole-exome analyses.

We also rectified the term in the Abstract, which reads as follows:

Page 2, line 40: We adopted JG1 as the reference for confirmatory exome re-analyses of seven Japanese families with rare diseases and found that re-analysis using JG1 reduced **total candidate** variant calls versus GRCh37 while retaining disease-causing variants.

8. Line 364, I think it is premature to use “several advantages” here. The authors didn’t

present “several advantages” of JG1 as a new reference. Please rephrase here or put more examples of the claimed advantages.

We rephrased the sentence, which reads as follows:

Page 18, line 474: Moreover, we demonstrated that JG1 exhibits **an advantage** as an ethnicity-matched reference, **at least** for NGS analyses within the clinical context of whole exomes of Japanese samples.

9. Line 395, a phased genome is better for characterizing the variants, which can increase nearly 30% SV calls than a un-phased genome (especially for heterozygous variants). It would be better if the authors can conduct phasing.

We agree that the phased diploid assembly would be more comprehensive in cataloguing SVs and enhance the representativeness of the major alleles. However, because a phased diploid assembly requires additional data and intensive computation, we are planning to present that analysis as a separate study. Indeed, construction of three sets of phased assemblies for the three subjects based on independent computation from this study has been in progress using the most up-to-date versions of FALCON, FALCON-Unzip, and FALCON-Phase software (pb-assembly), with additional Hi-C data, and we plan to present the results as a separate work. The **Review-only Table** below summarizes the SV detection with phased-assemblies and confirms that phasing increases SV detection by ~30%.

Moreover, we added a phrase stressing the importance of phased assembly for comprehensive detection of SVs in the Discussion section:

Page 20, line 518: Because the two haploid genomes can be regarded as a random sample from a panmictic population, assembling two haploid genomes per individual can increase the representativeness of variations—**especially structural variations**—in the population.

Review-only Table

sample	assembly	type	Insertions	Deletions	Total
jg1a	jg1a.draft*	un-phased haploid	7,859	5,039	12,898
	jg1a.phase0 [†]	phased haploid	8,202	5,160	13,362
	jg1a.phase1 [†]	phased haploid	8,077	5,133	13,210

	jg1a.phased0 and 1	diploid**	10,507	6,838	17,345
jg1b	jg1b.draft*	un-phased haploid	7,994	4,914	12,908
	jg1b.phase0†	phased haploid	8,204	5,124	13,328
	jg1b.phase1†	phased haploid	8,196	5,056	13,252
	jg1b.phased0 and 1	diploid**	10,584	6,737	17,321
jg1c	jg1c.draft*	un-phased haploid	7,893	4,992	12,885
	jg1c.phase0†	phased haploid	8,069	5,122	13,191
	jg1c.phase1†	phased haploid	8,078	5,123	13,201
	jg1c.phased0 and 1	diploid**	10,454	6,870	17,324

* The polished assemblies provided in this study

† Phased haploid assemblies constructed from the same PacBio data as in this study with additional Hi-C reads (HindIII-digested) using the pb-assembly software.

** Merged SV calls of the two phased haploids.

Note: The SVs were detected by genome-by-genome alignment with nucmer (v4.0.0beta2) software with '--maxmatch -l 100 -c 500 -t 24' option against GRCh38, followed by variant calling with Assemblytics (v1.0) software with '10000 (unique_length required parameter)' option. After converting the assemblytics output to VCF by SURVIVOR (v1.0.6) software convertAssemblytics command, SVs were counted and the two haploid-based VCFs were merged by SURVIVOR merge command with '1000 1 1 1 0 50' options.

Responses to Reviewer #2:

We sincerely appreciate the reviewer's helpful comments and evaluation. We addressed all of the concerns raised by the reviewer. The reviewer's comments are written in blue, whereas our responses are in black, and revised text in the manuscript is red.

Reviewer #2 (Remarks to the Author):

The authors first describe the construction of JG1 reference genome from de novo assembly of PacBio long reads combined with Bionano Genomics' optical maps, followed by pseudo-molecule construction based on genetic and radiation hybrid maps, and then demonstrate the utility of JG1 as a Japanese reference genome. This work is well organized and very comprehensive.

Again, we appreciate the evaluation that our work is well organized and comprehensive.

Here're minor comments:

1. The 2nd sentence from line 47 went wrong.

We agree that the original sentence was not correct, and we therefore rectified it. The revised sentence reads as follows:

Page 2, line 49: The complete genome sequence—also called "the reference genome"—is currently used as a target for mapping the enormous number of short reads generated using major next-generation sequencing (NGS) techniques^{3,4}.

2. The exact name of the company is Bionano Genomics. Therefore, its full name should be given first at line 110 as done for Pacific Bioscience at line 109 for consistency.

We corrected the company name. The revised text reads as follows:

Page 5, line 112: ... for example, Pacific Biosciences (PacBio) single molecule, real-time (SMRT) long reads (~10 kb in length) and Bionano Genomics (Bionano) optical mapping, which generates a high-resolution physical map^{20,23–25}.

3. As long as I know, Bionano raw data are called optical maps but their *de novo* assembled maps are called genome maps, which are used for hybrid scaffolding with PacBio contigs.

We corrected the notations; (1) "raw data" and "raw molecules" were changed to "optical maps" and (2) "optical maps" signifying assembled maps were changed to "genome maps". The corrected sentences are as follows:

Page 5, line 116: 2) Bionano **optical maps** are also *de novo* assembled (independent of the PacBio assembly) to yield **genome** maps; and 3) the PacBio-derived contigs are scaffolded by the Bionano **genome** maps.

Page 7, line 183: We also obtained deep Bionano **optical maps** for each subject (123× and 140× for two enzymes for jg1a; 160× and 175× for one enzyme for jg1b and jg1c, respectively; Supplementary Fig. 5 and Supplementary Table 2) and performed *de novo* assemblies of these **optical maps** to generate **genome** maps (Supplementary Table 5). Each *de novo* assembly of the Bionano **optical maps** was performed in two rounds (rough and full) to guarantee independence relative to the GRC reference genome (see Methods section). We then performed hybrid scaffolding between the PacBio-derived contigs and the Bionano-derived **genome** maps.

Page 24, line 622: We obtained two sets of Bionano **optical maps** using two different enzymes, Nt.BspQI and Nb.BssSI, for subject jg1a, and one set of Bionano **optical maps** was obtained with DLE-1 for jg1b and jg1c. In both cases, the Bionano **optical maps** were assembled in two steps—a rough assembly step and a full assembly step—to perform *de novo* assembly as independently as possible from the reference.

Page 45, line 1131: Pink boxes denote N-gaps, which are unresolved regions linked by Bionano **genome** maps, or the putative PAR1 region in the Y chromosome.

We also corrected the notations in **Supplementary Figs. 3 and 5** and **Supplementary Table 2**.

4. The meaning of the sentence starting with “Although assemblies” at line 117 is a bit confusing. In fact, many good *de novo* assemblies are not organized into pseudo-molecules limiting their utility as a reference genome. In this sense, I think it’s better to

use “pseudo-molecules” instead of “a set of chromosomes”.

We added the phrase "pseudo-molecules" after "a set of chromosomes". The corrected text reads as follows:

Page 5, line 121: Although assemblies generated in recent studies were highly contiguous and accurate, the assembled sequences were rarely anchored to a set of chromosomes (i.e. pseudo-molecules), thus making their use as references for NGS analyses impractical.

Page 5, line 126: ... a Korean reference genome was constructed by *de novo* assembly of the genome sequence of a Korean individual, reconstructed as pseudo-molecules, and rare variants were substituted with short reads from 40 Korean individuals.

5. Around line 269, at first glance, it appears to claim that JG1 is NOT localized within the Asian cluster but associated with the Japanese cluster. This might have been caused by uncertainty of PCA interpretation due to small variances assigned to the major two components. So the variances explained should be given in the PCA plot.

We agree that the variances explained by the two components were small, and we therefore added the proportion of variance explained by each PC in **Figure 2a** and **2b**. In addition, we provided PCA plots of PC1 versus PC3 and PC2 versus PC3 in new **Supplementary Fig. 9** to show some of the remaining variance.

We also modified the sentence describing the PCA analysis of JG1 in the Results section for more clarity, as follows:

Page 12, line 302: Notably, the JG1 haplotype localized near but outside of the Asian cluster; it localized to the most distant site both from the European and African populations, ...

6. In line 315, ‘piling-up’ is not an indication of an insertion but of a copy gain. It’s difficult to detect an insertion with unique sequence, purely based on read depth.

We agree that piling-up does not indicate the insertions but does indicate copy gains. We therefore corrected the sentence, which reads as follows:

Page 15, line 380: As shown in Figure 3c, **copy-number gains, representing some of the detected insertions**, were typically associated with a 'piling-up' of the average depth, ...

7. The description of Mate-pair sequencing from line 450 doesn't make sense. The sizes of mate pair libraries are 500bp on average?

We agree that the original description of the mate-pair sequencing was confusing. Actually, "tagmented" genomic DNA in the size of 2–15 kb (genomic DNA fragmented by a transposome) was circularized without size selection. Then, the circularized DNA was further fragmented by ultra-sonication, and the biotinylated mate-pair fragments (300–1000 bp) were purified and used for library preparation. Finally, the obtained libraries were size selected to 300–800 bp (peak at 500 bp) using AMPure XP beads. To clarify these points, we revised the text, which reads as follows:

Page 22, line 581: *Mate-pair sequencing*: Genomic DNA from nucleated blood cells was used for library construction with a Nextera Mate Pair Library Preparation kit (Illumina), according to the manufacturer's gel-free protocol, which produces a broader range of fragment sizes (2–15 kb). The obtained libraries were size-selected to 300–800 bp (peak at 500 bp) using AMPure XP beads (Beckman Coulter; Indianapolis, IN) and sequenced on a HiSeq 2500 system (Illumina) with a TruSeq Rapid PE Cluster kit (Illumina), and TruSeq Rapid SBS kit (Illumina) to obtain 201-bp paired-end reads.

Moreover, we provided the length distribution of the genomic DNA fragments estimated from the genomic interval of properly oriented read pairs as revised **Supplementary Fig. 7**.

8. It is unclear why "amplification" should be used when it's just electronic PCR at line 576. Please clarify the meaning.

We rephrased the term to "*in silico* amplification" to indicate that it is not an experiment. We changed the notation in the text, which reads as follows:

Page 28, line 721: *Detection of in silico STS marker amplification by electronic PCR*: We detected *in silico* amplification of the STS markers in the three genetic and six RH maps ...

Page 9, line 238: We searched for *in silico* amplification of STS markers by electronic PCR analysis of the meta-scaffolds and used ALLMAPS software³⁷ to order and orient the meta-scaffolds to build chromosomes.

Reviewer #1 (Remarks to the Author):

In the revised version, the authors addressed most of my concerns, and there are several minor issues that need to be taken care of.

1. For genome annotation of JG1, the authors annotated JG1 by lifting over GENCODE annotations based on the GRCh38 coordinates. I don't think it is a proper choice for annotating a de novo assembly genome. Given there are some assembly-specific sequences between JG1 and GRCh38, especially the filled gaps in GRCh38 by JG1 because these regions will not be covered in the liftover process and annotation. A de novo prediction method is recommended.

2. For evaluation of NRNR (non-reference non-repetitive), although the authors added this part in the revised manuscript, I don't understand why an old reference genome (hs37d5) was used instead of GRCh38? All NRNR study, including the original one (Kehr et al., 2017), used the latest version of reference genome to check NRNR.

3. In Table 1, some terms need to be adjusted. First, "Number of fragment" is not a good definition for evaluating an assembly. Does it mean the scaffold number or the final pieces of a genome? Use scaffold or contig numbers should be more clear. Second, the authors presented two N50 index: "N50 and NG50". What are the difference between these two indexes? Again, scaffold and contig N50 should be presented here.

Reviewer #2 (Remarks to the Author):

I still have one question regarding the PCA plot in Fig. 2. The fact that variances explained from the first two components in (a) are much larger than those in (b) shows that JG1 is indeed distinct from Northern Asian populations including JPT. What happens if the three basis genomes, jg1a, jg1b, and jg1c were included in (a)? Do they fall inside Northern Asian cluster? The authors need to explain why this happens even when they chose three typical Japanese genomes as a basis for JG1.

Responses to Reviewer #1:

We again sincerely appreciate the reviewer's helpful comments. We have addressed all of the concerns raised by the reviewer. The reviewer's comments are written in blue, whereas our responses are in black, and the revised text in the manuscript is red.

Reviewer #1 (Remarks to the Author):

In the revised version, the authors addressed most of my concerns, and there are several minor issues that need to be taken care of.

Again, we thank the reviewer's evaluation that we addressed most of the reviewer's concerns.

1. For genome annotation of JG1, the authors annotated JG1 by lifting over GENCODE annotations based on the GRCh38 coordinates. I don't think it is a proper choice for annotating a *de novo* assembly genome. Given there are some assembly-specific sequences between JG1 and GRCh38, especially the filled gaps in GRCh38 by JG1 because these regions will not be covered in the liftover process and annotation. A *de novo* prediction method is recommended.

The reviewer recommended a *de novo* gene prediction on JG1 and especially on the sequences that filled the gaps of GRCh38. JG1 has 36 distinct regions with 7,317 bp in total that uniquely filled 36 gaps in GRCh38. According to the reviewer, we performed *de novo* gene prediction on these regions using the state-of-the-art *de novo* gene prediction software AUGUSTUS. However, no genes were found in these regions. Besides, we performed *de novo* gene prediction on the newly described 164 non-reference non-repetitive (NRNR) sequences in JG1 (discussed below) and found that 20 regions harbored predicted genic regions (Supplementary Table 17). Moreover, we performed *de novo* gene prediction throughout the JG1 genome sequence and found 34,868 predicted genes. We summarized the results in a new Supplementary Note, Supplementary Fig. 9, and Supplementary Table 10. We added sentences that describe these new results as follows:

Page 11, line 281: Moreover, we assessed whether JG1 can fill the remaining gaps in the reference GRCh38²⁴, and found that JG1 uniquely filled 36 gaps, which was the second-highest number of uniquely filled gaps among other assemblies

(Supplementary Table 9). The 36 gap-filling sequences in JG1 did not apparently have genic regions (Supplementary Note; see also Supplementary Fig. 9 and Supplementary Table 10).

Supplementary Note: Supplementary Note 2: Evaluation of *de novo* gene prediction

We performed *de novo* gene predictions by AUGUSTUS software for the main chromosomes of GRCh38 (chromosomes 1–22, X, Y, and M) and compared the results with the GENCODE ver. 29 dataset (Supplementary Fig. 9). We considered the longest transcript per protein-coding gene from GENCODE because AUGUSTUS outputs one transcript per one gene. AUGUSTUS predicted more genes than the GENCODE dataset, primarily because AUGUSTUS predicted the larger number of short transcripts. Comparing the two datasets demonstrated that AUGUSTUS predict the larger number of gene regions with exon-level sensitivity higher than 60% (Supplementary Table 10). We observed similar results for JG1. These results suggested that AUGUSTUS can predict candidate novel genes with a modest, if not very high, sensitivity. However, no protein-coding genes were predicted for the 36 putative novel sequences that filled the GRCh38 gaps. Therefore, these 36 regions either do not encode any protein-coding genes or the gene prediction software is still not sensitive enough to detect possible genes.

2. For evaluation of NRNR (non-reference non-repetitive), although the authors added this part in the revised manuscript, I don't understand why an old reference genome (hs37d5) was used instead of GRCh38? All NRNR study, including the original one (Kehr et al., 2017), used the latest version of reference genome to check NRNR.

According to the reviewer's suggestion, we newly evaluated NRNR by using GRCh38 as the reference. We described the new results as follows:

Page 15, line 391: The reference genome might lack some population-specific sequences^{13,14}, thus might make some short reads from samples in the population unmapped. To determine whether JG1 has Japanese population-specific sequences not present in the reference genome, we collected the unmapped reads of 1,070 Japanese individuals⁴⁴ when mapped to the reference GRCh38 and re-mapped them to JG1. Of the $581 \pm 21 \times 10^6$ reads per individual, $4.5 \pm 1.5 \times 10^6$ reads were flagged as

unmapped to GRCh38 ($n = 1,070$; mean \pm SD). Of these, we found that $98,670 \pm 11,798$ reads could be successfully mapped to JG1 with high mapping quality (MAPQ ≥ 20). These reads were mapped to 449,549 distinct regions in JG1, and 164 regions had at least one mapped read from every individual (Supplementary Table 17). Among the 164 regions, 128 exhibited similarity to previously reported non-reference sequences^{13,14}. Among the other 36 regions, 34 exhibited 89–100% identity to a previously reported human genome sequences with matched chromosomal origin if known (26/34 regions; Supplementary Table 18)⁴⁵. One of the remaining 2 regions was 3 kb in length and exhibited 88.7% identity for approximately only half of the region (1,469 bp) with a glucoside xylosyltransferase 1 from *Pan paniscus* (XM_014347211.2). The other region, which was 1.5 kb in length and found in chromosome 7, exhibited only 85.5% identity to a chimpanzee BAC clone CH251-285G22 sequence (AC190217.3) from a matched chromosome.

3. In Table 1, some terms need to be adjusted. First, “Number of fragment” is not a good definition for evaluating an assembly. Does it mean the scaffold number or the final pieces of a genome? Use scaffold or contig numbers should be more clear. Second, the authors presented two N50 index: “N50 and NG50”. What are the difference between these two indexes? Again, scaffold and contig N50 should be presented here.

We revised Table 1, Supplementary Tables 4 and 6, which now describe assembly statistics by scaffold and contig numbers separately in newly provided columns. Also, we omitted NG50 value, which indicates the cumulative length sum of scaffolds or contigs with length longer than the value exceeds the length of the reference genome (GRCh38 in this case), because (1) the value was not largely different from N50, which is more widely used than NG50, and because (2) the reviewer had suggested at the first review round to simplify the Tables.

Reviewer #2 (Remarks to the Author):

We again sincerely appreciate the reviewer's insightful comment. We have addressed the concern raised by the reviewer. The reviewer's comment is written in blue, whereas our response is in black, and the revised text in the manuscript is red.

I still have one question regarding the PCA plot in Fig. 2. The fact that variances explained from the first two components in (a) are much larger than those in (b) shows that JG1 is indeed distinct from Northern Asian populations including JPT. What happens if the three basis genomes, jg1a, jg1b, and jg1c were included in (a)? Do they fall inside Northern Asian cluster? The authors need to explain why this happens even when they chose three typical Japanese genomes as a basis for JG1.

The reviewer questioned why JG1 was plotted outside the Northern Asian populations in the PCA plot with world-wide populations even when the three basis genomes jg1a, jg1b, and jg1c were constructed from typical Japanese samples. The possible reason why JG1 was plotted outside the Asian cluster but at the most distant site from European or African populations is that JG1 has the major allele among the Japanese population on most SNP sites throughout the genome due to the majority decision process. The majority decision process removed minor alleles in the Japanese population that can be major in African or European populations. Because of this effect, JG1 was plotted most distant from African or European populations than any other Asian individual, including Japanese and hence, the outside of the Asian cluster, although it might seem to be counterintuitive. To confirm this effect, we made a haploid genome by replacing minor alleles in the reference genome (hs37d5) with major allele among the Japanese population throughout the SNP marker sites and performed PCA with this "mock JG1" genome (Supplementary Fig. 12). The mock-JG1 was plotted outside the Asian cluster and most distant site both from European and African populations when the mock JG1 genome has the major allele of about $AF \geq 0.6$. Besides, we confirmed that the jg1a, jg1b, and jg1c genomes were included in the Asian cluster (in the new Supplementary Fig. 11d–f). On the other hand, including jg1a, jg1b, and jg1c along with JG1 in the Fig. 2a should be inappropriate and make the resulting plot uninterpretable because they are artificially "related" to JG1.

Although the PCA results were impressive, we agree that it may be somewhat counterintuitive and confusing. Therefore we decided to move the PCA plot from Fig. 2a

to Supplementary Fig. 11a and discussed the analyses in a newly provided Supplementary Note.

The new Fig. 2a now includes a PCA plot of JG1 with Japanese haplotypes (Fig. 2a). The revised paragraph now reads as follows:

Page 11, line 288: To assess whether JG1 is a representative reflection of the SNV composition of the Japanese population, we performed PCA **with several settings** using JG1, the reference GRCh38, 13 assemblies from diverse populations, **and haplotypes constructed from 11 HapMap3 populations** (see Methods section and Supplementary Tables 6 and 9 for the derived population of the assemblies). **First, we performed PCA with JG1 and 172 Japanese (JPT) haplotypes and confirmed that JG1 was plotted within the Japanese cluster (Figure 2a; see Supplementary Fig. 10 for plots with PC3). Second, we performed PCA with JG1, 5 other Asian assemblies, and 506 Asian haplotypes constructed from three Asian populations: JPT, Han Chinese (CHB), and Chinese in Denver (CHD) (Figure 2b). The PCA plot included two distinct clusters (namely, Japanese and others), with the JG1 haplotype associated with the Japanese cluster. Third, we performed PCA with JG1, GRCh38, and world-wide populations, and found that the JG1 haplotype localized near the cluster of Asian populations, whereas the GRCh38 haplotype localized between the African and European populations, as expected, based on the donors' ancestries (Supplementary Fig. 11a). Notably, the JG1 haplotype localized near but outside of the Asian cluster; it localized to the most distant site both from the European and African populations than any other Asian haplotypes, suggesting an "Asianness" when compared with the other two populations. We reasoned that this occurred because JG1 harbors the major allele among the Japanese population in most SNP sites due to the majority decision procedure that removes the minor allele among the Japanese populations. The removed minor allele among the Japanese population can be the major in European or African populations. That would be why JG1 was plotted most distant from the two populations than any Asian haplotype (for further discussion, see Supplementary Note, Supplementary Fig. 11 and 12).**

Page 47, line 1154: Figure 2. SNV characteristics of JG1. **(a)** PCA plot of the haplotype SNP composition of JG1 and HapMap3 Japanese in Tokyo (JPT) samples.

Supplementary Note: Supplementary Note 3: PCA of JG1 haplotype

Why was JG1 plotted outside the Asian cluster? The plotted location of JG1 was indeed outside the Asian cluster, but more importantly, the position was in a more distant site both from the European and African clusters. This pattern means that JG1 was more distant from European and African populations than any haplotype in the Asian population, i.e., the genetic composition of JG1 was more distant from the two populations. This was confirmed by the fact that JG1 was plotted at the most distant site from the European or African populations when we performed PCA using Asian and African (Supplementary Fig. 11b) or Asian and European populations (Supplementary Fig. 11c), respectively. On the other hand, the base assemblies *jpg1a*, *jpg1b*, and *jpg1c*, which were used to construct JG1, were plotted in the Japanese population (Supplementary Fig. 11d–f).

Then, why was JG1 located away from both European and African populations? JG1 adopted the major allele among the three assemblies *jpg1a*, *jpg1b*, and *jpg1c* in its construction process. This majority decision resulted in adopting the major allele in the Japanese population at most SNP sites (Fig. 2c). The major allele in a Japanese population is often minor in European or African populations and vice versa. Therefore, JG1 can be considered more non-European and non-African than any Japanese haplotype or even any Asian haplotype. For these reasons, JG1 was located distant from both European and African clusters in the PCA.

To gain further insights into the effect of this adoption of the major allele on PCA plot location, we generated "mock JG1" haplotypes that replaced the reference genome *hs37d5* with the major allele (non-reference allele frequency 50%–90%) in the Japanese population and performed PCA (Supplementary Fig. 12). We found that replacing the GRC-type allele with an allele harbored by $\geq 50\%$ or 60% of the Japanese haplotypes plotted outside the Asian cluster and distant from European and African populations just like JG1 (Supplementary Fig. 12).

Reviewer #2 (Remarks to the Author):

I really appreciate the authors' efforts to answer my question regarding the location of JG1 on the PCA plot. The authors' explanation with the added figures and notes will be very helpful for those who are interested in population specific reference genomes.

Response to Reviewer #2:

The reviewer's comments are written in blue, whereas our responses are in black.

Reviewer #2 (Remarks to the Author):

I really appreciate the authors' efforts to answer my question regarding the location of JG1 on the PCA plot. The authors' explanation with the added figures and notes will be very helpful for those who are interested in population specific reference genomes.

We would like to thank the reviewer's comments and evaluation of our revision.